# Climate drivers and palaeobiogeography of lagerpetids and early pterosaurs

Davide Foffa [1,2,3] ✉, Emma M. Dunne [4] ✉, Alfio Alessandro Chiarenza [5] ✉, Brenen M. Wynd [6], Alexander Farnsworth [7,8], Daniel J. Lunt [7], Paul J. Valdes [7,8], Sterling J. Nesbitt [3], Ben T. Kligman [9,10], Adam D. Marsh[10], William G. Parker [10], Richard J. Butler [2], Nicholas C. Fraser [1,11], Stephen L. Brusatte [1,11] & Paul M. Barrett [12]

The origin of pterosaurs, the first vertebrates to achieve powered flight, is poorly understood, owing to the temporal and morphological gaps that separate them from their closest non-flying relatives, the lagerpetids. Although both groups coexisted during the Late Triassic, their limited sympatry is currently unexplained, indicating that ecological partitioning, potentially linked to palaeoclimate, influenced their early evolution. Here we analysed pterosauromorph (pterosaur + lagerpetid) palaeobiogeography using phylogeny-based probabilistic methods and integrating fossil occurrences with palaeoclimate data. Our results reveal distinct climatic preferences and dispersal histories: lagerpetids tolerated a broader range of conditions, including arid belts, enabling a widespread distribution from the Middle to early Late Triassic. Conversely, pterosaurs preferred wetter environments, resulting in a patchier geographical distribution that expanded only as humidity increased in the Late Triassic, probably following the Carnian Pluvial Event. This major environmental disturbance, potentially driven by changes in $CO_2$-related thermal constraints and/or palaeogeography, appears to have had a key role in shaping early pterosauromorph evolution by promoting spatial segregation and distinct climatic niche occupation.

Pterosaurs were the first vertebrates to evolve powered flight, more than 60 million years before the earliest birds. However, our understanding of early pterosaur evolution is hindered by the major temporal and anatomical gaps between these highly modified flying reptiles and their closest terrestrial relatives[1,2]. The fossil record of pterosaurs and their kin is notoriously incomplete[3–11] (Supplementary Fig. 1), and fundamental aspects of their early evolution, such as the timing, area, ecological settings of their initial radiation and palaeobiology (for example, growth dynamics and climate preferences[1,12,13]), are still poorly understood compared with other contemporaneous archosaur groups such as dinosaurs[14–16].

Recent phylogenetic studies, descriptions of new taxa and detailed reevaluations of historically described specimens have renewed interest in pterosaur origin, ancestry and early evolution. This work overturned previous hypotheses of pterosaur relationships, demonstrating that lagerpetids were the closest relatives of pterosaurs, uniting these two groups in the clade Pterosauromorpha[2,11,17–19]. All known lagerpetids are non-flighted forms, whereas all known pterosaurs are volant. This new phylogenetic hypothesis substantially reduces temporal and anatomical gaps between Pterosauria and other avemetatarsalians (pterosaurs, dinosaurs and their closest relatives)[2,11,18,19]. Consequently, the origin of Pterosauria is now minimally constrained to an interval of ~10–15 million years between the Ladinian and early Norian intervals of the Triassic Period[2,11,18,20]. The past decade of research has also provided a wider sample of Triassic pterosauromorph specimens from more localities across the globe and a better understanding of their phylogenetic relationships[1,2,11,17–20] and stratigraphic occurrences. These advances offer the chance to investigate early pterosauromorph macroevolution

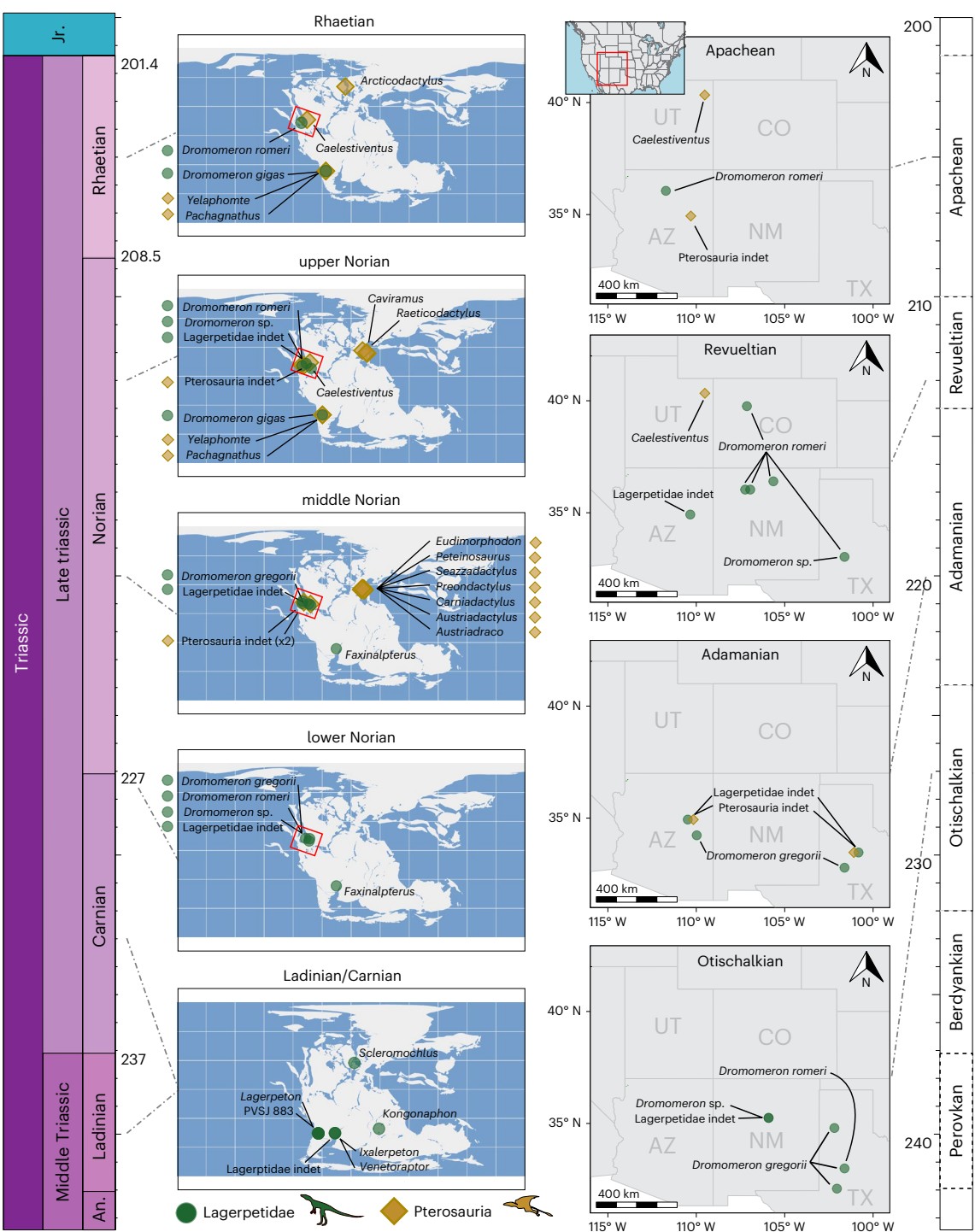

**Fig. 1 | Middle–Late Triassic pterosauromorph occurrences worldwide (left) and close-ups of the southwestern USA (right).** An., Anisian; Jr., Jurassic.

quantitatively (for example, ref. 11), by uncovering potential abiotic factors that might have influenced their diversification, as hinted at by the stratigraphic occurrence and disjunct palaeogeographical distributions of lagerpetids and pterosaurs through the Triassic[18].

Sympatric occurrences of lagerpetids and pterosaurs were limited during their ~20-million-year overlap in the Norian and Rhaetian (Fig. 1). Lagerpetids are found in continental fluviolacustrine deposits on at least four current continents and maintained a wide latitudinal and geographic spread throughout most of their evolutionary history[11,17,18,20–34] (late Ladinian to late Rhaetian, 237–201 Ma; Fig. 1 and Supplementary Fig. 1). Conversely, the earliest body fossils of unambiguous pterosaurs

are middle Norian in age and found in a restricted low-latitude belt, primarily, but not exclusively, in marine formations that were deposited around the margins of Tethys[1,12,35–37]. From the late Norian onwards, pterosaurs spread to higher latitudes and in a much broader array of habitats[1,3,8,12,38–42]. This could suggest that lagerpetids and pterosaurs occupied fundamentally different ecological, environmental or climatic niches. Investigating pterosauromorph distribution in time and space combined with environmental data allows a new opportunity to answer several long-standing key questions in palaeontology: how did Triassic climate change affect pterosaur and avemetatarsalian evolution? What climatic regimes and environments did vertebrate

flight first evolve in? And why are early pterosaur fossils so rare, and where might we find more of them?

We investigated these questions using a holistic approach that integrates novel fossil occurrence data, comprehensively sampled phylogenies, climatic niche modelling and palaeobiogeography. Our aims are (1) to assess and quantify the potential for latitudinal dispersal of Triassic lagerpetids and pterosaurs and to compare it with that of other contemporaneous groups; (2) to investigate climatic preferences of pterosauromorphs; and (3) to quantify and map habitat suitability for pterosaurs and their closest relatives during the Middle to Late Triassic. Furthermore, our results allow us to estimate the ancestral area of origin and hypothetical distribution and dispersal of pterosauromorphs across Pangaea during the Middle Triassic to early Late Triassic—the undersampled time interval in which their origin probably unfolded—to propose testable hypotheses regarding their early biogeography and its potential influence on their evolution and to identify potential target areas for future fieldwork.

## Results

### Palaeobiogeography
Our palaeobiogeographic analysis suggests that lagerpetids—and pterosauromorphs as a whole—probably originated in southwestern Pangaea (that is, modern South America), whereas the origin of pterosaurs was predicted at low latitudes in the Northern Hemisphere (Extended Data Fig. 1), consistent with previous studies[8,11]. Note that this and the following results are based on phylogenetic data, which, in light of the biases and incompleteness of the fossil record may simplify what are probably more complex scenarios (Supplementary Fig. 1).

**Potential of latitudinal dispersal.** The pterosauromorph dispersal inertia curve maintains a consistently high Δ-log likelihood (that is, lower but constant dispersal) across the Ladinian–early Norian interval (Fig. 2a). In other words, the dispersal of lagerpetids (the only pterosauromorphs present at that time) across climatic barriers was constant but more constrained than that of dinosaurs (and avemetatarsalians as a whole). Dinosaurs experienced relatively stronger barriers to dispersal in the Ladinian–early Carnian and late Norian–Rhaetian, interrupted by a 'release phase' (that is, increased levels of dispersal) during the middle–late Carnian[14] (Fig. 2a). The middle Norian–Rhaetian segment of the pterosauromorph curve shows an increase in Δ-log likelihood, potentially indicating a reduced crossing of barriers through this interval (Fig. 2a). This section of the curve is primarily, if not exclusively, influenced by the addition of pterosaurs (Methods). The pattern of reduced dispersal is surprising given that Triassic pterosaurs were volant and many are found in low-latitude areas (Fig. 1, Extended Data Fig. 1 and Supplementary Fig. 1). A similar pattern (that is, increase in species richness and drop in dispersal) could be achieved by few intermittent crossings of the barriers, which are unlikely to alter the dispersal pattern, followed by cladogenesis within the same geographical area, which will not affect the dispersal through barriers. Nevertheless, as emphasized in the Methods, the importance of this segment of the curve remains uncertain. As pterosaurs could fly and were presumably less constrained by physical barriers, their likelihood for dispersal may have left a less clear phylogenetic signature compared with their non-flying close relatives.

**Accumulated latitudinal dispersion.** A complementary way to decipher the latitudinal dispersion of different groups is by quantifying dispersal events in each clade through time[11] (Fig. 2b,c). This shows that lagerpetids dispersed consistently throughout their evolutionary history (Anisian–Rhaetian). The lagerpetid curve for accumulated latitudinal dispersion is high both in absolute (Fig. 2b) and corrected values (Fig. 2c). These curves maintain a plateau throughout the Carnian and Norian and decrease only in the Rhaetian, when the taxonomic diversity of the group declines. This pattern is similar to that of

silesaurids (Discussion), contrary to the results of Müller et al.[11], whose lagerpetid curve peaked after the Carnian and was more similar to that of dinosaurs. This inconsistency is caused by (1) the pruning by Müller et al.[11] of key lagerpetids (*Kongonaphon kely* and PVSJ 883) from their dataset, which reduced the number of branches and dispersal events in the Anisian–Carnian interval, thus lowering that part of the curve; and (2) the different strategy of Müller et al.[11] of incorporating uncertain biogeographical values at internal nodes, which led them to discard the dispersal events at these particular nodes. The accumulated latitudinal dispersal of pterosaurs peaks immediately after their appearance in the middle Norian and maintains a high profile throughout the Norian and Rhaetian[11] (Fig. 2b,c). The abrupt and pronounced Norian peak in the pterosaur curve may underscore the inherent ease of dispersal for these flying reptiles compared with their land-bound counterparts.

### Palaeoclimate niche occupation
Statistical comparisons reveal that pterosaurs and lagerpetids occupied different climatic niche spaces during their temporal overlap in the Late Triassic (Fig. 3 and Extended Data Table 1; pairwise permutation multivariate analysis of variance (MANOVA), $P = 0.0127$). During the Norian and Rhaetian, lagerpetids occupied areas characterized by warmer temperatures and drier conditions and with less pronounced seasonal excursions in temperatures (Fig. 3b–d). Comparatively, pterosaurs occurred in areas that were distinctly cooler and seasonally more variable (Fig. 3b,d and Extended Data Table 1; $P < 0.01$, $P < 0.05$). Both lagerpetids and pterosaurs occupied areas with similar values of mean annual precipitation (MAP; Fig. 3c and Extended Data Tables 1 and 2; $P > 0.05$), although a majority of lagerpetids occupied drier areas (Fig. 3c). Pterosaurs occurred across a wide range of values for annual precipitation and seasonal variation in precipitation (Fig. 3c,e). The palaeoclimate niche occupation of lagerpetids was not constant over time (Fig. 3f). Specifically, Ladinian and Carnian lagerpetids (that is, *Kongonaphon, Ixalerpeton, Lagerpeton* and PVSJ 883) are found in high-latitude areas in southern Gondwana that were on average colder and withstood higher seasonal fluctuations in temperatures than their Norian and Rhaetian counterparts (Fig. 3 and Extended Data Table 1; pairwise permutation MANOVA, $P = 0.012$). This pattern is not entirely surprising considering the palaeogeographical occurrence of Ladinian and Carnian lagerpetids compared with their younger counterparts. Most Norian–Rhaetian occurrences are, in fact, found at lower latitudes that were characterized by higher temperatures and smaller seasonal fluctuations in temperature (Fig. 3b,d and Extended Data Table 1; $P < 0.001$), with the exception of *Faxinalipterus* and *Dromomeron gigas*, which were found in the southern part of Pangaea, in geographical areas and palaeoclimatic regimes broadly similar to those of their pre-Norian predecessors (Discussion, Figs. 1 and 3f and Supplementary Fig. 1). As a result, lagerpetids as a whole occupied an overlapping (pairwise permutation MANOVA, $P = 0.098$; Extended Data Fig. 2 and Extended Data Table 1) but broader palaeoclimate niche than pterosaurs, probably related to their higher tolerance for warmer, more seasonal (temperature-wise) and tendentially drier conditions (Extended Data Fig. 2 and Extended Data Tables 1 and 2).

### Habitat suitability modelling
Climatic suitability offers a route to interpret the palaeobiogeographical distributions of early pterosauromorph body fossils and allows us to identify areas of suitable habitable space in fossil-depleted intervals. It provides a coarse, but useful, representation of their hypothetical fundamental niches (Fig. 3) in geographic space (Fig. 4 and Extended Data Fig. 3). The temporal and spatial biases that control the early pterosaur fossil record epitomize the challenges faced in obtaining a comprehensive understanding of their evolutionary history and geographic distribution (Supplementary Fig. 1). Nevertheless, a probabilistic macroecological approach can provide insights on their climatic preferences and potential habitable areas, predicting potential geographic

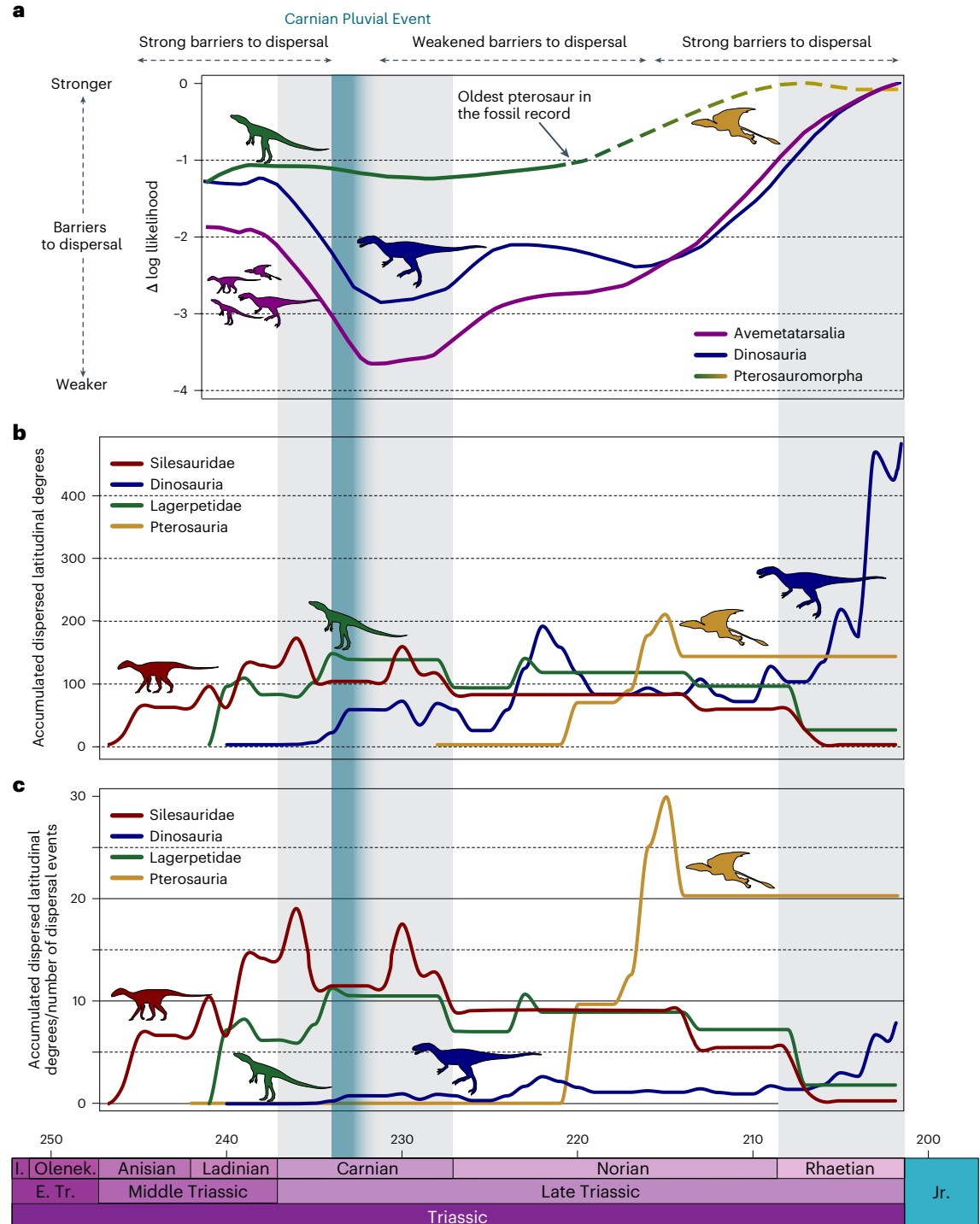

**Fig. 2 | Palaeobiogeograpic analyses of Middle–Late Triasssic pterosauromorphs. a**, A plot showing the potential of dispersal of avemetatarsalian groups through the Triassic (based on Griffin et al.[14]). **b**, Absolute accumulated degrees of latitudinal dispersal of avemetatarsalian groups. **c**, Event-corrected accumulated degrees of latitudinal dispersal of avemetatarsalian groups. J., Jurassic; E. Tr., Early Triassic; I., Induan; Olenek., Olenekian.

ranges for pterosaurs and lagerpetids based on climatic suitability. The distribution of areas of high suitability of pterosaurs and lagerperids appear broadly complementary. However, pterosaur potential distribution is patchier overall with lower habitability in continental interior areas (that is, arid regions) compared with the more widely distributed of lagerpetids. Although pre-Norian pterosaur body fossils have yet to be recovered, ecological niche modelling indicates the presence of potential suitable habitable space for hypothetical early pterosaurs

that was unevenly spread at low tropical latitudes during the Ladinian (Fig. 4 and Extended Data Fig. 3), compared with the occupation of higher, more temperate latitudes in the Northern Hemisphere and broader and more continuous suitable space occupation in the southwestern tropics, for lagerpetids (Fig. 4). Habitat suitability modelling also indicates a marked extension of habitable pterosaur space towards the coastlines of Northern Tethys in the Carnian–early Norian (Fig. 4), alongside a reduction of suitable continental zones, emphasizing their

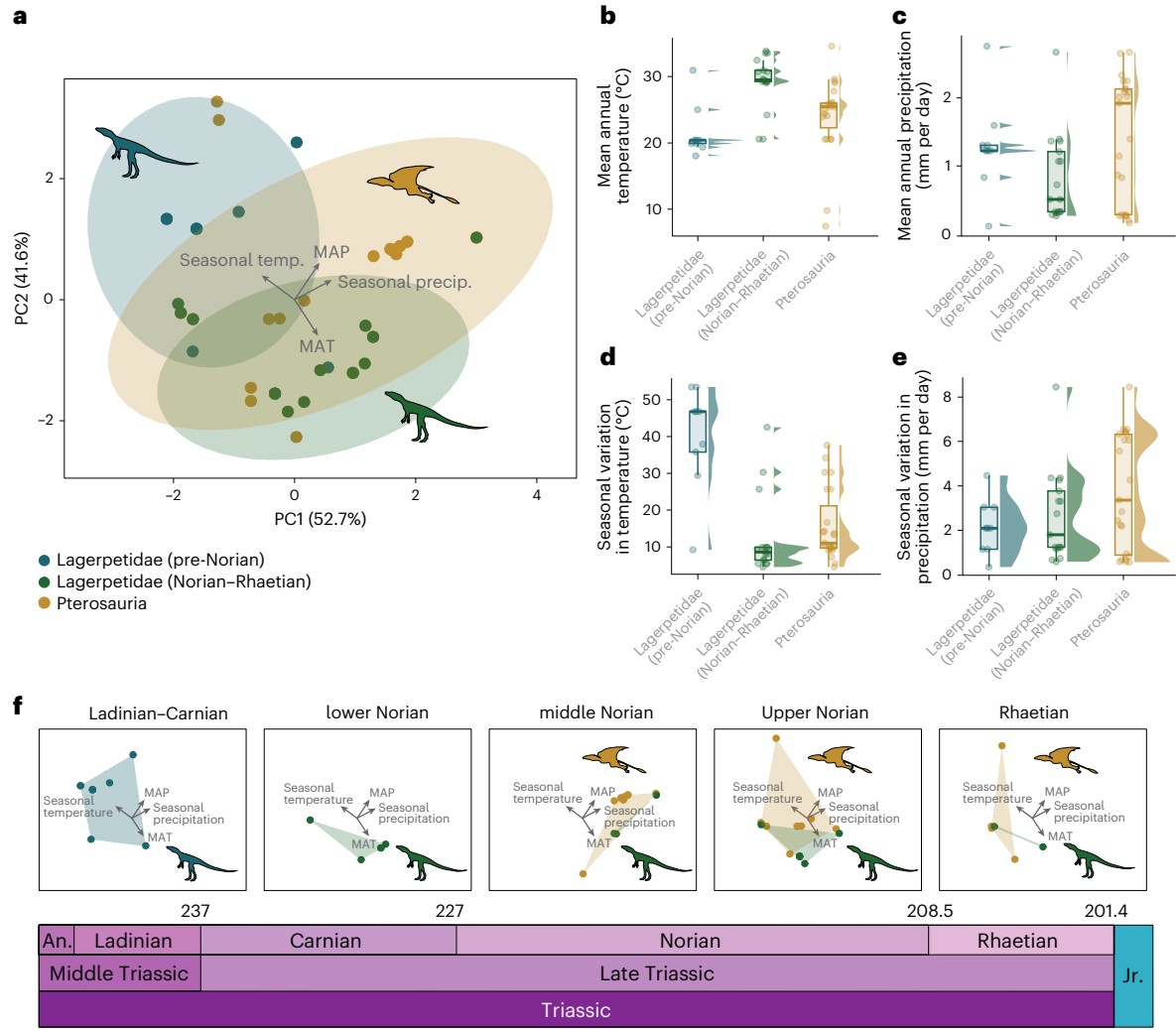

**Fig. 3 | Climatic preferences of Middle–Late Triassic pterosauromorph lineages. a**, Principal component analysis (PCA) of variables of climatic importance showing pterosaurs ($n = 23$), pre-Norian lagerpetids ($n = 9$) and Norian–Rhaetian lagerpetids ($n = 17$). **b**–**e**, Raincloud plots comparing climatic variables between pterosaur and lagerpetids: mean annual temperature (**b**); mean annual precipitation (**c**); seasonal variation in temperature (**d**); seasonal variation in precipitation (**e**). **f**, The time-sliced PCA space occupation of lagerpetids and pterosaurs through the Triassic. Results of statistical comparisons are reported in Extended Data Table 1. P, precipitation; T, temperature. The box plots display the median and interquartile range (IQR), the upper whiskers extends from the 75th percentile to the largest value and the lower whiskers extends from the 25th percentile to the lowest value.

climatic preference for the more temperate and wetter environments (Extended Data Figs. 3 and 4). By contrast, lagerpetids display a distinct Carnian pattern, maintaining a more homogeneous latitudinal occupation but with decreasing habitat suitability along the southern margin of the northern Tethys gulf, while simultaneously exhibiting an increasingly higher preference towards the southwestern tropics and reaching slightly higher latitudes in the Southern Hemisphere (Fig. 4). During the Norian (our 'training stage'; Fig. 4 and Extended Data Fig. 3), pterosaurs (now represented by body fossils) were widely distributed in coastal, wet, tropical and low-latitude areas (Figs. 1 and 4). This is accompanied by the continuing patterns of reduction in suitable continental interior areas, except alongside the palaeoequatorial belt (broadly corresponding to the Chinle and Dockum basins) where favourable conditions for the group are maintained. Overall, in the Norian and Rhaetian, areas of pterosaur high climatic suitability occur at broad latitudes, extending from the northern margin of the Tethys to northern Australia and central South America (that is, southern Brazil and northern Argentina). Low suitability is recorded at high latitudes outside the tropics and in what is now equatorial Africa, Antarctica, Oceania and northeastern Eurasia and continental interiors

(for example, modern Brazil and Canada–USA border). This pattern is broadly maintained through the late Norian and Rhaetian with an expansion of high suitable zones along the eastern margin of Pangaea (Fig. 4 and Extended Data Figs. 3 and 4). By contrast, lagerpetid suitability habitats are maintained throughout the Norian–Rhaetian interval with the addition of the western palaeoequatorial belt (that is, Chinle and Dockum basins) and northwestern Africa.

## Discussion

### Palaeobiogeography, palaeoclimate and ecological niches

Triassic pterosauromorph distribution suggests that pterosaurs and lagerpetids had distinct and divergent early evolutionary histories, with regard to biogeography and dispersal. Lagerpetids achieved a wide latitudinal distribution soon after their first appearance (predominantly in southern Pangaea) and maintained a presence at high and low latitudes, globally, throughout their evolutionary history[10,11,18] (Fig. 1 and Supplementary Fig. 1). The clade's dispersal across climatic barriers remained low and constant through time (Fig. 2), with our results suggesting that lagerpetids had a broad tolerance for a range of different climatic conditions, for example, higher thermal tolerances and/or other behavioural

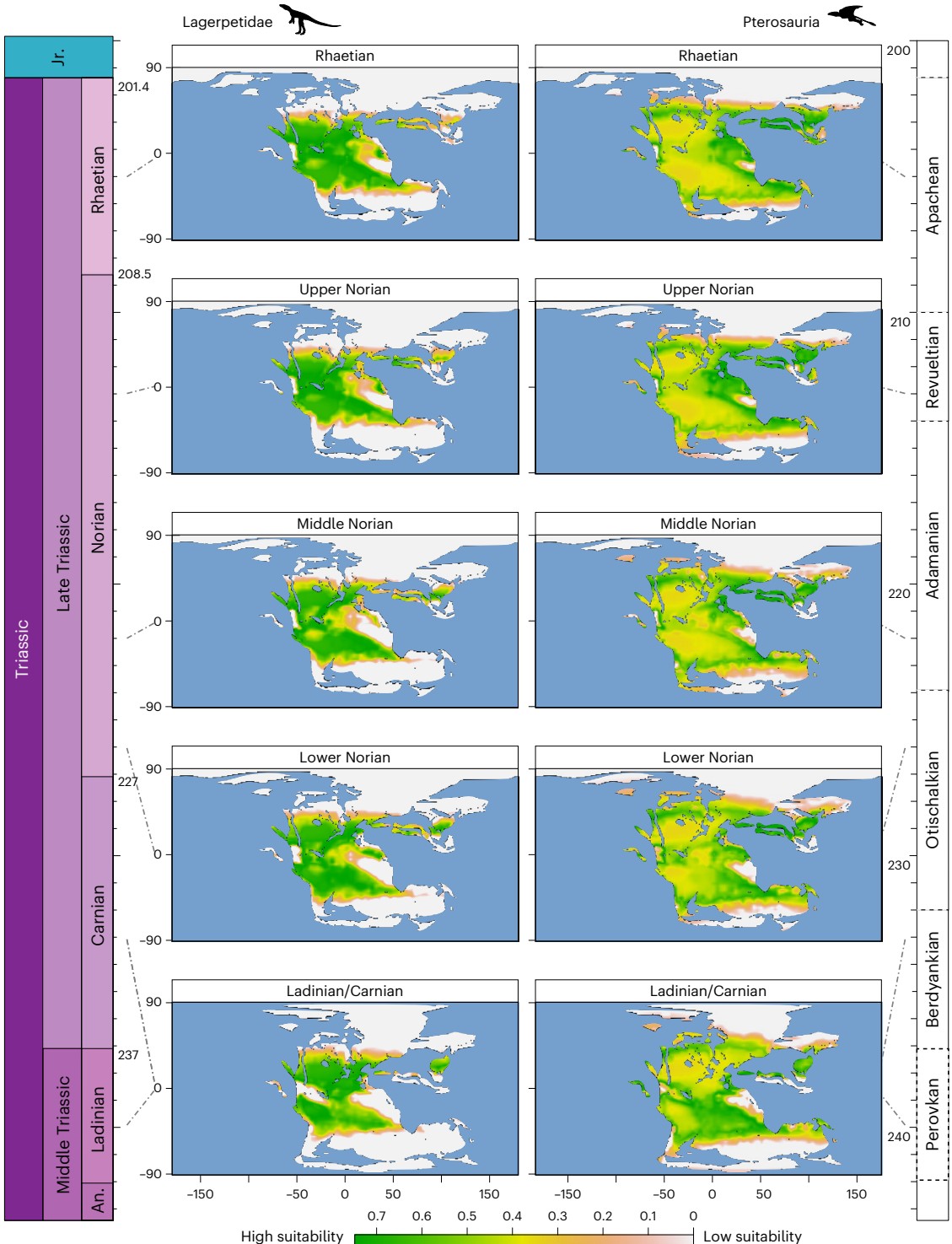

**Fig. 4 | Climatic suitability of early pterosaurs and lagerpetids through the Ladinian–Rhaetian interval.** Note that the pre-Rhaetian suitability maps are modelled predictions because, so far, there is no Ladinian–Carnian fossil record of pterosaurs.

strategies to deal with high temperatures. By contrast, the earliest pterosaurs are found in low-latitude lacustrine and fluvial beds of the Chinle and Dockum basins during the early and middle Norian and more predominantly (but not exclusively) in coastal habitats of the northern Tethyan region[3] (Fig. 1 and Supplementary Fig. 1). However, within a few million years, pterosaurs are then found worldwide in remarkable variety of environments including many beyond low-latitudinal coastal regions, as shown by *Arcticodactylus* from fluvial deposits in Greenland (Fleming Fjord Formation, Norian–Rhaetian)[38], *Caelestiventus* from the

deserts of southwestern USA (Nugget Sandstone, upper Norian–Hettangian)[41] and *Pachagnathus* and *Yelaphomte* from Argentina (Quebrada del Barro Formation, upper Norian–lower Rhaetian)[20]. In line with this clade's dramatic peak of dispersal in the middle Norian (Fig. 2), it is possible that these younger pterosaur occurrences are evidence of the clade's first dispersals outside of its hypothesized low-latitude ancestral area. As such, these occurrences might represent successive invasions of diverse habitats following a climatically driven ecological release (see below), rather than evidence of ancestral preference. The

hypothesis of a rapid global spread[8] from low latitudes finds support in (1) the phylogenetic predominance of low-latitude taxa closer to the root of the clade (2) and the ~6-million-year gap between the earliest identifiable pterosaurs in the fossil record (that is, SMU 69125, and PEFO 45782 from the early Norian of the southwestern USA) and the first occurrences of pterosaurs outside the palaeotropics (that is, *Pachagnathus*, *Yelaphomte* and *Arcticodactylus* from Greenland and South America).

The stratigraphic and biogeographical distribution of lagerpetids and pterosaurs, and any other group, is the result of prominent interaction of biotic and abiotic factors. The palaeoclimatically informed phylogeographic model and spatially explicit ecological niche models we generated offer an explanation for the above-mentioned biogeographical patterns. First, our climate-based principal component analysis (PCA) shows that, in the Norian–Rhaetian, lagerpetids and pterosaurs had statistically distinct climate preferences (Fig. 3a and Extended Data Table 1). Lagerpetids occupied a broad climatic niche and are found to inhabit pronoucedly warmer (Fig. 3b and Extended Data Table 1) and drier (Fig. 3c and Extended Data Table 1) localities than those yielding Triassic pterosaurs. The occurrence of lagerpetids under varied climatic conditions suggests a remarkable ecological versatility, which may have enabled their more continuous distribution across different latitudes (Fig. 4). By contrast, early pterosaurs exhibit a patchier distribution, particularly in terms of mean annual and seasonal temperatures (Fig. 3d,e). Pterosaur distribution may indicate a narrower, more specialized climatic preference and/or that these suitable conditions were initially limited to equatorial areas. Nevertheless, suitable conditions for pterosaurs are found globally, although in discontinuous patches, and less frequently in continental interiors, in the late Norian–Rhaetian (Fig. 4b,d), in accordance with the pattern of quantified latitudinal dispersal and increased geographical spread in the fossil record (Fig. 1). Another possibility is that, during this early stage of pterosaur evolutionary history, perhaps owing to initially constrained flight capabilities and/or partial confinement to arboreal environments[1] (for example, to protect from predation and thermal stress or abundant food supply), their biogeography was constrained by the distribution of the canopy habitat they inhabited. As more suitable areas became available by the Norian (Fig. 4), pterosaurs may have dispersed latitudinally, benefiting from potentially increased suitable habitats. However, their climatic preferences might have resulted in a sparse and locally restricted distribution, a possibility confirmed by their modelled, patchily distributed suitable areas. This aspect could potentially be attributed to their limited capabilities for long-distance aerial dispersal and/or a stage that was still partially confined to arboreal environments. This unique ecology could account for the relatively high dispersal inertia observed in pterosaur lineages (Fig. 2a) and more pronounced latitudinal dispersion towards the latest Triassic (Fig. 2b,c) (that is, endemic dispersion within climatic blocks). Our combined analyses suggest that changes in the distribution and extension of climatically suitable areas throughout the Triassic are congruent with latitudinal dispersal patterns, demonstrating limited geographical overlap between lagerpetids and pterosaurs. This may exemplify the intricate interplay between climatic conditions and habitat, as well as the development of diverse pterosauromorph ecological niches, that probably acted on the spatiotemporal distribution of this clade. Paradoxically, pterosaurs survived the end-Triassic mass extinction but lagerpetids did not, despite pterosaurs occupying a patchier, narrower climatic niche and having a more restricted geographic distribution than lagerpetids before this event. This suggests that other factors such as the ability to fly might have played important roles in the differential survivorship of these two clades during this extinction.

## On the origin of pterosaurs

Historically, the evolution of powered flight in vertebrates (that is, pterosaurs and bats[43,44]) has resulted in a geologically instantaneous and geographically widespread distribution; however, until recently, a lack of early pterosaur fossils prevented any assessment into whether the evolution of a volant vertebrate leads to a predictable biogeographic outcome. Now, efforts to understand the timing and place of pterosaur origin have increased in concert with recent discoveries. A literal reading of the Triassic pterosaurs record suggests that the group originated at low latitudes in northern Pangaea. This hypothesis is supported both by the early-diverging phylogenetic positions of European taxa[8] and by the oldest occurrence of indeterminate specimens of this group in the early–middle part of the Norian in the southwestern USA. However, it is worth noting that, while there is probably a genuine biogeographical signal in these observations, they could also reflect severe sampling biases[4–6]. We can now explore this issue by combining the results of our analyses to provide a potential model that incorporates the biogeographical and climatic circumstances around this event. However, we must first address the absence of Middle Triassic–early Late Triassic pterosaurs from the fossil record.

The most recent divergence time estimates based on pterosauromorph phylogenetic relationships constrain pterosaur origin to a 10–15-million-year interval that extends across the Ladinian–early Norian[2]. However, while an Early Triassic or Middle Triassic origin for pterosaurs cannot be excluded, no body or trace fossils are known before the Norian[1,8,12]. Most of the earliest pterosaurs have been found in black shale lithologies in mid-latitudes from the middle Norian Alpine arch (that is, Italy, Austria and Germany). Similar facies are found in the same geographical area throughout the Anisian–Carnian interval, but despite the high potential for exceptional preservation at these localities, pre-Norian pterosaurs have yet to be found (or are currently unrecognized) from well-sampled Middle Triassic and early Late Triassic Lagerstätten not only in this region (for example, Monte San Giorgio, late Anisian–late Ladinian, Italy and Switzerland; upper Buntsandstein, Muschelkalk and lower Keuper, Anisian–Carnian, in Germany and equivalent beds in central Europe; Lunz, early Carnian, Austria) but also worldwide (Yangjuan, Anisian, China; Digxiao, Ladinian, China; Xinpu, Carnian, China; Raibl, early Carnian, Italy; Madygan, Ladinian–Carnian, Kyrgyzstan)[9,12], even after decades of extensive sampling. Similarly, the absence of pterosaurs remains unexplained in the lower part of the Newark Supergroup in North America (that is, Lockatong Formation and Cow Branch Formation, lower Norian), which has yielded the fragile remains of gliding reptiles such as *Icarosaurus* and *Mecistotrachelos*, among others[45,46].

Ecological niche modelling offers a potential explanation for pterosaur absences in these localities. Our analyses suggest that many of these areas were climatically unsuitable for pterosaurs before the Norian (Fig. 4). For instance, the northern margin of the Tethys Ocean was climatically poorly suitable for pterosaurs until the late Carnian (Fig. 4). This could support the idea that pterosaurs originated shortly before their first appearance in the fossil record and/or that pterosaurs originated earlier, and elsewhere, but did not have a global distribution until later in the Norian[1,12]. However, while our model does not necessarily support a pre-Norian origin of the group in the Tethyan region, it provides an additional explanation for their earlier absence from this area and time interval. Specifically, our results suggest that there is potential for pre-Norian and early-Norian pterosaurs to be found elsewhere, in areas identified as habitable in our palaeoclimatically informed suitability models (Fig. 4 and Extended Data Fig. 3). Our niche models suggest that it might have been possible for pterosaurs to occur in the late Middle Triassic (Anisian–Ladinian) deposits of the southwest USA (upper Moenkopi Formation of Arizona and New Mexico), Morocco (members T3–T4 of the Timezgadiouine Formation, Argana Basin), India (Yerrapalli Formation), China (Xingyi fauna), Tanzania (Lifua Member of the Manda Beds), Brazil (lower Santa Maria Supersequence) and potentially southern and central Europe (for example, Dont Formation, Italy; Vellberg Formation, Germany). Suitable areas during the Carnian include the southwestern USA (for example, lower

Otischalkian beds in the Dockum Group of Texas and New Mexico), Brazil (middle Santa Maria Formation), Argentina (Chañares Formation and lower Ischigualasto Formation), India (Tiki Formation and Maleri Formation), Madagascar (Isalo II/Makay Formation), continental Europe (Krasiejów locality, Poland), the lower Norian beds of the southwestern USA (upper Otischalkian and lowermost Adamanian beds of the Chinle Formation and Dockum Group, Arizona, Texas and New Mexico) and the basins of the Newark Supergroup, among others. In addition to these areas, the South China block shows high climatic suitability throughout the Anisian–early Norian. The upper portion of the Anisian–Carnian Guanling Formation (Panxian Fauna, Guanling Biota) also has high climatic suitability to yield pterosaur fossils, although the fossil record of its marine fauna shows limited terrestrial influence.

These results carry the caveat of known biases in the pterosaur fossil record[4–6,9] (Supplementary Information) but provide testable hypotheses for future targeted fieldwork. While these formations were deposited in areas of high climatic suitability for pterosaurs, their geological characteristics might not have been suitable for preserving their fossils. As noted previously, many Triassic pterosaurs are preferentially preserved in black shale units[1,12], although partial skeletons and individual bones have been more commonly recovered from fluvial, alluvial and desert depositional environments more recently[20,38,41]. Accordingly, among the above-mentioned formations, we particularly emphasize target lithologies representative of palaeoenvironments with known high preservation potential, particularly microvertebrate bonebeds, as these facies have the highest chance to preserve the small, fragile body fossils of Triassic pterosaurs. Articulated or associated skeletons comprise nearly all Triassic pterosaur occurrences, suggesting that their record in deposits where bones are recovered as disassociated and isolated elements (for example, microvertebrate bonebeds) may be underrepresented due to a lack of recognizable diagnostic features in collected skeletal elements from these settings.

## Conclusions

Our study examines the historical palaeobiogeography of pterosaurs and their close relatives, lagerpetids, which together comprise pterosauromorphs. We identify climatic limits shaping the distribution of Triassic pterosauromorphs. Although multiple factors probably contributed to the distribution of these groups, the broad climatic adaptability and wide latitudinal spread of lagerpetids suggest ecological versatility, which allowed members of this group to thrive in localities with considerably warmer conditions than those yielding Triassic pterosaur remains. By contrast, early pterosaurs had a more constrained distribution in terms of mean annual and seasonal temperatures, favouring wetter areas that became more common after the Carnian, enabling their spread during the latest Triassic.

Our study independently aligns with the hypothesis suggesting that pterosaurs became widespread soon after their likely origin in the Middle Triassic–Late Triassic[2]. Furthermore, the modelled Carnian–early Norian pterosaur climatic niche would probably be found in temperate and wet conditions typical of tropical, near-coastal low-to-middle-latitude depositional environments. In emphasizing the complex interplay between climatic conditions, habitat and the development of diverse ecological niches, our study contributes valuable insights into the evolution and distribution of pterosaurs and their close relatives.

## Methods

### Palaeobiogeography

To explore the phylo-palaeobiogeography of Triassic pterosauromorphs, we implemented complementary analyses based on two recently published studies to (1) assess the degree to which pterosauromorphs dispersed across climatic barriers compared with other avemetatarsalian clades (following ref. 14) and (2) quantify the latitudinal dispersal of lagerpetids and pterosaurs alongside that of other avemetatarsalians (following ref. 11). To better compare the results of these analyses, which drew on different datasets, we assembled and tip-dated an avemetatarsalian supertree (Extended Data Fig. 1). This grafts the dinosaur tree topology recovered by ref. 14 into a randomly selected most parsimonious tree of Archosauromorpha from ref. 11. We then manually incorporated the dinosaurian and silesaurid taxa present in the latter dataset but not the former. This strategy allowed us to retain the entire pterosauromorph sample included in ref. 11, which is the most comprehensive available for this group. Each taxon in this newly assembled dataset was then scored for two different sets of geographical areas (see below). This was deemed necessary because the two above-mentioned studies were designed using distinct geographical frameworks (see details below). Overall, these operations improved the sampling of the selected avemetatarsalian groups and allowed more meaningful comparisons between the two sets of biogeographical analyses. Our analyses cover the Ladinian–Rhaetian interval, but the taxonomic sample includes Jurassic species, enabling consideration of all potential dispersal events nested in the Triassic.

Griffin et al.[14] demonstrated the limited impact of changing archosaur topology in their results. However, their tree did not include a broad sample of lagerpetids and pterosaurs, so here we focus on the impact that different pterosauromorph tree topologies could have on our results. In Pterosauromorpha, the relationships within Pterosauria are relatively well established, so the topology here adopted corresponds to the strict consensus from ref. 11 and derivative analyses. Conversely, the relationships within Lagerpetidae are not well resolved, and it is not possible to produce a fully resolved strict consensus topology. However, for the purpose of our biogeographical analyses, this does not matter because the alternance of 'Laurasian' and 'Gondwanan' areas of origin at the tips is broadly maintained: a South American taxon (that is, *Faxinalipterus*) is consistently found at the earliest diverging branch, followed by *Scleromochlus*, from Europe. Further up the tree, a minority of North America taxa (that is, *Dromomeron romeri* and *Dromomeron gregorii*) are found nested within a majority of South American species. Our choice of using a randomly selected tree following ref. 14 is thus justified because the patterns of geographical areas at the tips of the tree are broadly stable and the patterns of potential of latitudinal dispersal and accumulated latitudinal dispersal are not affected by the taxon order of the tips (Supplementary Information).

**Potential of latitudinal dispersal.** First, we implemented an event-based quantitative analysis that aimed to assess the potential for pterosauromorph latitudinal dispersal during the waxing and waning of climatic barriers, following the methodology of ref. 14. Griffin et al.[14] originally developed this method to test "whether the early phylogenetic history of dinosaurs retains a signal for restricted dispersal" (p. 317). Here, we applied the protocol to pterosauromorphs, which were sparsely represented in the latter study. We limit our interpretation of the pterosauromorph curve to the Ladinian–Carnian portion of the whole interval, where the signal is driven primarily by lagerpetids, because we cannot be sure to what extent our model is applicable to volant taxa, as pterosaurs affect the Carnian–Rhaetian portion of the results.

We implemented this analysis following the methodology of Griffin et al.[14]. First, we scored each taxon in the newly assembled dataset for one of five geographical regions: eastern Laurasia, western Laurasia, equatorial belt, northern Gondwana or southern Gondwana. These regions were selected because of the biogeographical importance of their boundaries throughout the Ladinian–Rhaetian (our interval of interest), namely, an arid belt in low-latitude southern Pangaea[47] and, less relevant to the time interval of our study, the Hispanic Corridor and Viking Strait[48,49]. Second, we extended the lower boundary of the analyses to include the Ladinian stage to include the complete stratigraphic range of lagerpetids. We then implemented a dispersal–extinction–cladogenesis (DEC) model on the time-calibrated phylogeny under maximum likelihood. To simulate the waxing and waning of

arid climatic barriers to dispersal, we adopted distinct rate matrices of dispersal across selected geographical areas and tested their occurrence across the whole interval. The Δ-likelihood is a measure of how strongly the dispersal pattern is supported at any time: in other words, a more negative Δ indicates dispersal/waning of climatic barriers. The Δ measures how much dispersal is occurring by imposing a penalty when the climate barrier is crossed. Thus, the Δ will be more negative when more organisms (lineages) are dispersing when barriers are present. Conversely, a Δ closer to zero indicates strong barriers to dispersal, or that the crossing of barriers was sporadic and largely followed by cladogenesis. Overall, this model uses the phylogeny of different avemetatarsalian groups to test whether, and when, climatic barriers affected the dispersal of each group, particularly lagerpetids, during the Ladinian–Rhaetian.

**Accumulated latitudinal dispersion.** We also quantified the latitudinal dispersal of each avemetatarsalian group, following the methodology of Müller et al.[11]. Differing from that study, we decided against pruning the lagerpetids *Kongonaphon kely* and PVSJ 883 to enhance completeness. The stratigraphic and palaeobiogeographical occurrences of these taxa are of paramount importance for our analyses, and we believe that the benefits of their inclusion outweigh the uncertainties in their phylogenetic positions. Specifically, *Kongonaphon kely* is not only the stratigraphically oldest lagerpetid but also the only member of this clade from this region of Gondwana (Madagascar, representing the Indian subcontinent + Madagascar continental area). Similarly, we retained the unnamed taxon PVSJ 883, from the late Carnian of Argentina[10,16], because its occurrence suggests a dispersal event that would otherwise be ignored.

To run the analyses, we used the same tip-dated supertree of Avemetatarsalia as discussed above. This time, each taxon in the dataset was assigned to one of eight discrete geographic areas (that is, western North America, eastern North America, Brazil, Argentina, Europe + Russia, South Africa, and Indian subcontinent + Madagascar) among the ten determined by Button et al.[50]. The quantification of dispersal events relies on reconstructing ancestral states at internal nodes. This was achieved using the R package BioGeoBEARS[51] v.1.1.2 in R v.4.3.1 (ref. 52), using different likelihood-based palaeobiogeographic models: DEC, dispersal vicariance analysis likelihood (LIKEDIVA) and BAYAREALIKE. Only the simple DEC model was used for subsequent calculations to avoid long-distance palaeolatitudinal jumps in the latitudinal dispersion calculations[14] following Müller et al.[11], and for meaningful comparison with the first analysis that is also based on DEC[14] for the same reason. However, we notice that, in our comparisons of biogeographical models, '+j' models are better supported (Supplementary Table 1). This may suggest a more complex biogeographic history in which (aerial?) dispersal rather than vicariance may have been the dominating phenomenon (expected in flying animals) in early pterosaur macroevolution.

Because the main barriers preventing dispersal in the Middle to Late Triassic were arid latitudinal belts at mid-to-low latitudes[47], we quantified the accumulated amount of latitudinal dispersal of lagerpetids and pterosaurs separately and compared them with those of other avemetatarsalian clades (silesaurids and dinosaurs).

First, we counted the number of dispersal events across the tree by identifying those branches with different geographical values at their nodes (or tips). The latitudes of well-known avemetatarsalian-bearing localities were adopted as proxies for the broader geographic area and used to calculate the latitudinal extent of dispersal events. The localities we used are the same as those in Müller et al.[11]: Otis Chalk Quarries (western North America, palaeolatitude 6.9° N); New Haven County (eastern North America, 16.2° N); Buriol site (Brazil, palaeolatitude 39.7° S); Quebrada del Puma site (Argentina, palaeolatitude 40.6° S); Lombardia (Europe + Russia, palaeolatitude 27.1° N); Free State (South Africa, 42.8° S); and Manda (Indian subcontinent, palaeolatitude

53.7° S). Unlike ref. 11, we did not discard uncertain nodes but instead averaged the latitudinal values of different areas when an overwhelming signal was not computed. Only events that started (but did not necessarily end) in the Triassic were included in the final analyses.

The total amount of latitudinal dispersion (measured in degrees) for each clade was then calculated and plotted across the Anisian–Rhaetian through 1-million-year-long bins (Fig. 2b) and subsequently averaged by the number of dispersal events in each interval (Fig. 2c). These calculations were done using a slight modification of the R codes provided by Müller et al.[11] (see 'Code availability' section).

**Palaeoclimate niche occupation**

To explore the palaeoclimatic niches occupied by pterosauromorph lineages, we compiled a locality-based dataset of all Triassic lagerpetid and pterosaur occurrences. This integrated a literature search with unpublished fieldwork data (20+ years in the southwestern USA in Arizona, Texas and New Mexico)[3,11,17,20,28–38,40,53–74] (see 'Dataset_pterosauromorphs_R1.xls' in Supplementary Data 1). Unique occurrence data were recorded for each specimen in our dataset. Unnamed and/or undescribed specimens were examined in person and included to increase the data available for our palaeoclimate niche analyses. This sampling strategy means that the datasets for the palaeobiogeographic and climate analyses are independent (that is, the first relies on phylogeny and the second on occurrence data), but all taxa present in the former are also included in the latter. This strategy recognizes the value of including undescribed or indeterminate lagerpetid and pterosaur specimens, which provide fundamental information on the distributions of their respective clades, even if they cannot be included in phylogenetic analyses. The lagerpetids included in the palaeoclimate niche analyses but excluded from the phylogenetic framework (and, hence, the palaeobiogeographic analyses) are: the indeterminate lagerpetids (NMMMNH P-80469, PEFO 44476 and PEFO 50545) from, respectively, NMMNH L-149 in the Los Esteros Member of the Santa Rosa Formation of New Mexico (Otischalkian holochronozone, lower Norian)[64], PFV 456 (Thunderstorm Ridge) in the Blue Mesa Member (Adamanian holochronozone, middle Norian)[34] and PFV 215 (Zuni Well Mound) in the Petrified Forest Member (Revueltian holochronozone, upper Norian)[34] of the Chinle Formation of Petrified Forest National Park and multiple representatives of the genus *Dromomeron* from 13 distinct localities in the southwestern USA[30,31,34,59–62,67] (Fig. 1 and Supplementary Data 1). Similarly, the pterosaurs that are present in the palaeoclimate analysis but could not be included in the phylogenetic dataset are *Arcticodactylus* from Greenland[38,40], which is currently under investigation[68]; three unnamed pterosaurs from the southwestern USA, including an undescribed pterosaur (tentatively referred to as *Eudimorphodon* sp.) from the 'Kalgary localities' (approximately upper Carnian–lower Norian) of the Tecovas Formation of the Dockum Group[3,69] and fig. 9.5 in ref. 70, two undescribed pterosaurs from the Chinle Formation of Petrified Forest National Park, one (PEFO 45782) from PFV 456 (Thunderstorm Ridge) in the Blue Mesa Member (Adamanian holochronozone, middle Norian, -220 Mya (refs. 73,74)); and an undescribed pterosaur (PEFO 53384) from the Owl Rock Member (Apachean holochronozone, late Norian)[71]. Finally, MCSNB 8950 is an unnamed taxon from the Argilliti di Riva di Solto Formation of northern Italy (upper Norian), which was previously referred to *Eudimorphodon ranzi*[72] but is now thought to be an unnamed new genus[12].

However, because the resolution of the general circulation (palaeoclimatic) models is 1° × 1° (-111 km²) (see below), we consider only one locality if more than one specimen was found in a radius smaller than 111 km, unless they yielded different taxa (for example, the pterosaur localities in northern Italy). The occurrence of taxa with uncertain stratigraphic ranges covering multiple time bins was considered present in each of the relevant bins. This resulted in a summarized dataset of 54 occurrences (24 pterosaur and 30 lagerpetid entries; Supplementary Data 1).

**Palaeoclimate reconstructions.** To explore whether palaeoclimatic conditions influenced the biogeographic distributions of pterosaurs and lagerpetids, we integrated occurrence data for these groups with outputs from a general circulation (palaeoclimate) model. The values extracted from these models show a good match with the estimates from geochemical, sedimentological and biological proxies at the basin level and broader scales (for example, see refs. 75,76 for a detailed study of the palaeoclimate of the Chañares–Los Rastros–Ischigualasto Triassic succession). The accuracy and evaluation of general circulation (palaeoclimate) model outputs are further discussed in the Supplementary Information. Palaeoclimate model simulations used a recent version of the coupled Atmosphere–Ocean General Circulation Model and HadCM3L[77] (specifically HadCM3L-M2.1aD), following the nomenclature in the work of Malanoski et al.[78]), including the modifications described by Judd et al.[79]. The model has a resolution of 3.75° longitude × 2.5° latitude in the atmosphere and ocean, with 19 hybrid levels in the atmosphere and 20 vertical levels in the ocean with equations solved on the Arakawa B-grid. Atmospheric subgrid scale processes such as convection are parameterized as they cannot be resolved at the resolution of the model. The ocean model is based on the model of Valdes et al.[80] and is a full primitive equation, three-dimensional model of the ocean. A second-order numerical scheme is used along with centred advection to remove nonlinear instabilities. Flux adjustments (such as artificial heat and salinity adjustments in the ocean model to prevent the model drifting to unrealistic values) are not required in this model[81], which is crucial for long palaeoclimate simulations. Sea ice is calculated on a zero-layer model with partial sea ice coverage possible, with a consistent salinity assumed for ice. Because geological data recording land surface vegetation for Triassic stages are uncertain and globally sparse, we use a version of the model that includes the dynamical vegetation model TRIFFID (Top-Down Representation of Interactive Foliage and Flora Including Dynamics) and land surface scheme MOSES 2.1a (ref. 82). TRIFFID predicts the distribution and properties of global vegetation based on plant functional types (PFTs), in the form of fractional coverage (and, thus, PFT coexistence) within a grid cell, and is, in turn, based on competition equations based on the climate tolerance of five PFTs. The HadCM3L model demonstrates high performance at reproducing the modern-day climate[79] and has been used for an array of pre-Quaternary palaeoclimate experiments[79,83,84]. Palaeoclimate experiments typically require on the order of hundreds of years to reach a near-surface quasi-equilibrium state (but many thousands of years for the deep ocean[79,80]) as well as true climate equilibrium. Relatively low-resolution global climate models such as HadCM3L are relatively computationally expensive, allowing near-fully equilibrated simulations of climate to be undertaken that would not be possible with higher-resolution, more complex models[85].

Seven model simulations with Ladinian (239.54 Ma), 'early' Carnian (233.6 Ma), 'late' Carnian (232 Ma), 'early' Norian (227 Ma), 'middle' Norian (222.4 Ma), 'late' Norian (217.8 Ma) and Rhaetian (204.9 Ma) were carried out using stage-specific boundary conditions (topography, bathymetry, solar luminosity, continental ice and partial pressure of $CO_2$ ($pCO_2$)). Stage-specific realistic carbon dioxide concentrations were chosen on the basis of proxy-$CO_2$ (Ladinian, 1,034 ppm; 'early' Carnian, 1,492 ppm; 'late' Carnian, 1,614 ppm; 'early' Norian, 2,059 ppm; 'middle' Norian, 1,810 ppm; 'late' Norian, 1,481 ppm; and Rhaetian, 1,503 ppm) reconstructions from ref. 86, and the solar constant was based on ref. 87. The orography and bathymetry were derived from palaeogeographic digital elevation models, produced by ref. 88 as part of the PALEOMAP project (see ref. 79 for more details). Each stage-specific digital elevation model is interpolated from a 1° × 1° grid onto the model 3.75° × 2.5° grid. Similarly, land ice is also transformed onto the model grid assuming a simple parabolic shape to estimate the ice sheet height (m). Surface soil conditions were set at a uniform medium loam everywhere because stage-specific soil parameters during the Triassic are not globally known. All other boundary conditions (such as orbital

parameters, aerosol concentrations and so on) are held constant at preindustrial values. The simulations were carried out for a total of over 10,000 years. By the end of the simulations, (1) the globally and volume-integrated annual mean ocean temperature trend is less than 1 °C per 1,000 years; (2) trends in surface air temperature are less than 0.3 °C per 1,000 years; and (3) net energy balance at the top of the atmosphere, averaged over a 100-year period at the end of the simulation, is less than 0.25 W m$^{-2}$. Climatological means were produced from the last 100 years of each simulation. All these simulations are identical to the 'Scotese07' simulations described in ref. 78 and the 'model 2' simulations of ref. 79. The climate predicted by the model is compared with proxy indicators of climate in the Supplementary Information. The outputs from the simulations are available at https://www.paleo.bristol.ac.uk/ummodel/scripts/html_bridge/scotese_07.html.

**Palaeoclimatic niche space.** To explore the climatic conditions occupied by pterosauromorphs during the Late Triassic, multivariate statistical tests were used and summary statistic plots were constructed. This approach is commonly used in modern ecology[89–93] and, in the past decade, has become more common in studies pertaining to the fossil record of both invertebrate[94–99] and vertebrate groups[15,16,73,100]. Accordingly, each taxon was assigned climate variables based on the mean values for their stratigraphic age and geographic locations (namely, mean annual temperature (MAT), MAP, seasonal variation in temperature and seasonal variation in precipitation; Fig. 3 and Supplementary Table 2). Taxa spanning two geological stages ($n = 8$) were assigned the mean of the variables for both stages and assigned single, averaged, stratigraphic occurrence. To quantify 'palaeoclimatic niche space' for each taxon, we followed the procedure outlined in ref. 16. Obtaining information on species' fundamental niches from the fossil record is challenging: therefore, the term 'palaeoclimatic niche' as used here refers to an approximation of the realized climatic niche of the fossil taxa (that is, the set of climatic conditions occupied by a taxon[16]). A PCA was performed using the prcomp() function in R v.4.4.2 (ref. 52), which included the scaling argument so that variables were scaled to have unit variance before the analysis took place. A non-parametric MANOVA was performed to statistically compare the distribution of the two groups using the R package RVAideMemoire[89]. Raincloud plots displaying both box plots (depicting the distributions of the palaeoclimate data within the taxonomic groups) and frequency distributions were constructed to examine the range of individual palaeoclimatic conditions occupied by both groups. To statistically compare the distributions between both groups, pairwise comparisons were performed in R using Wilcoxon rank-sum (Mann–Whitney) tests. These specific statistical tests were chosen because the palaeoclimate variable data do not conform to the assumptions of a normal distribution, which was determined through probability plots (for example, quartile–quartile plots) and Shapiro–Wilk tests.

## Habitat suitability modelling
To assess the climatic suitability of these taxa, we implemented the DOMAIN algorithm[100] through the R package 'dismo'[101], which is an ecological niche modelling tool using the Gower distance metric to assess climatic suitability. The DOMAIN algorithm quantifies the disparity between the climatic conditions of map pixels and the nearest species observation within the $n$-dimensional environmental space, diverging from geographic proximity. Although this model is generally classified as a coarse niche modelling technique[102–105], it offers the advantage of straightforward implementation and minimal assumptions, which are necessary due to the paucity of the pterosauromorph fossil record. Given the coarse spatial and temporal resolution of our dataset, coupled with our exclusive focus on producing suitability maps based solely on climate, and the low number of occurrences in our datasets, we favoured this simplistic modelling approach over more recently introduced ecological niche and habitat suitability modelling techniques,

including those previously used by our team in prior works[101–105], where the associated limitations and assumptions are also addressed. We furthermore calibrated the model on a compound occurrence dataset that maximizes the geographic spread for each of the subclades, centring on the best sampled interval (the middle–late Norian, 218 Ma) and including palaeobiogeographic outliers from the Carnian (Lagerpetidae) and throughout the full duration of the Norian for pterosaurs, projecting these models into more refined climatic simulations (eight time slices from the Ladinian to the Rhaetian). MAT (°C) and MAP (mm per year) are the variables used for the habitat suitability modelling.

## Reporting summary

Further information on research design is available in the Nature Portfolio Reporting Summary linked to this article.

## Data availability

All datasets can be found alongside their respective R codes via Figshare at https://figshare.com/s/742a30f30deb4aefb60a (ref. 106), Supplementary Data 1; https://figshare.com/s/7ac05f57d35b3810a2c7 (ref. 107), potential for latitudinal dispersion; https://figshare.com/s/8cabc46601eb64f9b9f0 (ref. 108), accumulated latitudinal dispersion; https://figshare.com/s/6254a692186e79fa9060 (ref. 109), palaeoclimate niche occupation; and https://figshare.com/s/6600233d24db8f1d986e (ref. 110), habitat suitability modelling., climate modelling data repository (https://www.paleo.bristol.ac.uk/ummodel/scripts/papers/Foffa_etal_2025.html). Source data are provided with this paper.

## Code availability

All datasets can be found alongside their respective R codes via Figshare at https://figshare.com/s/7ac05f57d35b3810a2c7 (ref. 107), potential for latitudinal dispersion; https://figshare.com/s/8cabc46601eb64f9b9f0 (ref. 108), accumulated latitudinal dispersion; https://figshare.com/s/6254a692186e79fa9060 (ref. 109), palaeoclimate niche occupation; and https://figshare.com/s/6600233d24db8f1d986e (ref. 110), habitat suitability modelling.

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

## Acknowledgements

D.F. is funded by Marie Skłodowska-Curie Actions: Individual (Global) Fellowship (H2020-MSCA-IF-2020; grant 101022550) and by the Royal Commission for the Exhibition of 1851–Science Fellowship in the first stages of this work. E.M.D. acknowledges the support of the Emerging Talents Initiative (FAU). A.A.C. was funded by a European Research Council (ERC) Starting Grant under the European Union's Horizon 2020 Research and Innovation Programme (grant 947921 "Mapas" to Sara Varela) and by a Juan de la Cierva Formación 2020 Fellowship (grant FJC2020-044836-I) from the Ministry of Science and Innovation and the European Union Next Generation EU/PRTR and a Royal Society Newton International Fellowship (grant NIF\R1\231802). R.J.B., E.M.D., A.F. and D.J.F. acknowledge a Leverhulme Research Project Grant (RPG-2019-365). A.F. and P.J.V. acknowledge NERC grants NE/X015505/1 and NE/X018253/1. A.F. and D.J.F. acknowledge the Leverhulme grant (grant no. RPG-2019-365). A.F. acknowledges NERC funding (grant NE/X013111/1) and the Chinese Academy of Sciences Visiting Professorship for Senior International Scientists (grant 2021FSE0001). S.J.N. was

supported by NSF EAR 1943286. This is Petrified Forest National Park Paleontological Contribution No. 97 and a contribution to the Natural History Museum's Evolution of Life Theme. The silhouettes in the figures are from PhyloPic (http://phylopic.org/) under Creative Commons licenses CC BY-NC-SA 3.0 and CC BY 3.0.

## Author contributions

D.F., E.M.D. and A.A.C. equally contributed to the design and execution of the project. D.F., E.M.D. and A.A.C. performed the analyses, created plots and figures and wrote the first draft of the manuscript. D.F. coordinated activities among all authors and led the reviews. D.J.L., P.J.V. and A.F. contributed to running the palaeoclimate simulations and climate analysis. A.F., D.J.F. and P.J.V. performed modelling experiments, evaluation and analysis. A.F. contributed to the first draft of the paper. B.M.W. assisted D.F. with the implementation of the biogeography analyses. B.T.K., A.D.M., W.G.P. and S.J.N. contributed to assembling the dataset with fieldwork data from the Chinle Formation and Dockum Group. P.M.B., R.J.B., N.C.F. and S.L.B. aided with the original design of the project and experiments and jointly supervised the work. All authors contributed to the writing and approval of the final paper.

## Competing interests

The authors declare no competing interests.

## Additional information

**Extended data** is available for this paper at https://doi.org/10.1038/s41559-025-02767-8.

**Correspondence and requests for materials** should be addressed to Davide Foffa, Emma M. Dunne or Alfio Alessandro Chiarenza.

[1]Department of Natural Sciences, National Museums Scotland, Edinburgh, UK. [2]School of Geography, Earth and Environmental Sciences, University of Birmingham, Edgbaston, UK. [3]Department of Geosciences, Virginia Tech, Blacksburg, VA, USA. [4]GeoZentrum Nordbayern, Friedrich-Alexander-Universität Erlangen-Nürnberg, Erlangen, Germany. [5]Department of Earth Sciences, University College London, London, UK. [6]Life Sciences, Spokane Falls Community College, Spokane, WA, USA. [7]School of Geographical Sciences, University of Bristol, Bristol, UK. [8]State Key Laboratory of Tibetan Plateau Earth System, Resources and Environment, Institute of Tibetan Plateau Research, Chinese Academy of Sciences, Beijing, China. [9]Department of Paleobiology, National Museum of Natural History, Smithsonian Institution, Washington, DC, USA. [10]Department of Science and Resource Management, Petrified Forest National Park, Petrified Forest, AZ, USA. [11]School of GeoSciences, Grant Institute, University of Edinburgh, Edinburgh, UK. [12]Fossil Reptiles, Amphibians and Birds Section, Natural History Museum, London, UK. ✉e-mail: davidefoffa@gmail.com; dunne.emma.m@gmail.com; a.chiarenza15@gmail.com

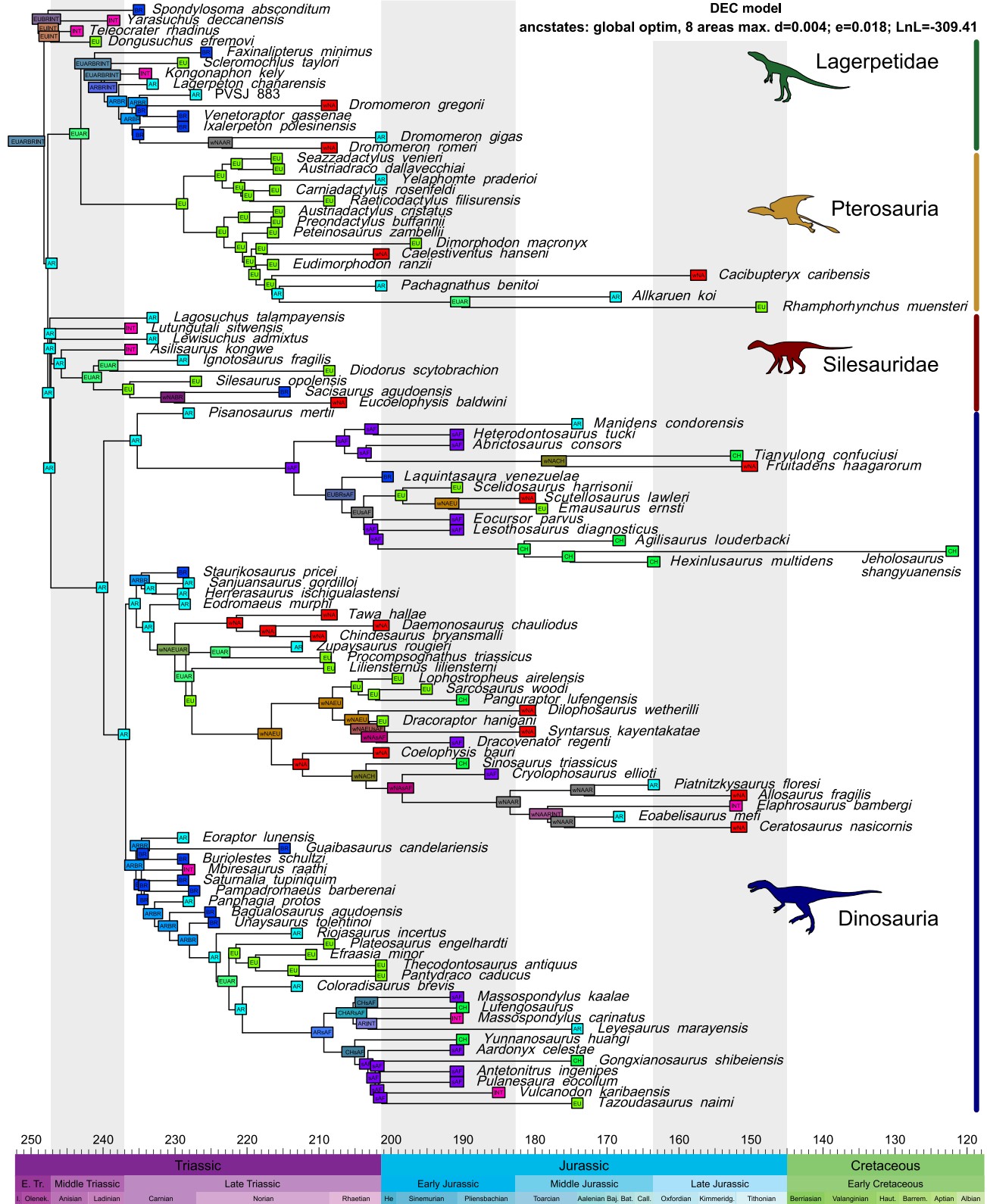

**Extended Data Fig. 1 | Time calibrated informal tree of early avemetatarsalian relationships and the results of DEC biogeography model.** Abbreviations: AR, Argentina; BR, Brazil, Uruguay, Namibia; CH, China, Thailand, Kyrgyzstan; EU, Europe, Russia and Greenland; INT, India, Tanzania, Zambia, Madagascar; wNA, western USA, British Columbia, Mexico and Venezuela; sAF, South Africa, Lesotho, Zimbabwe.

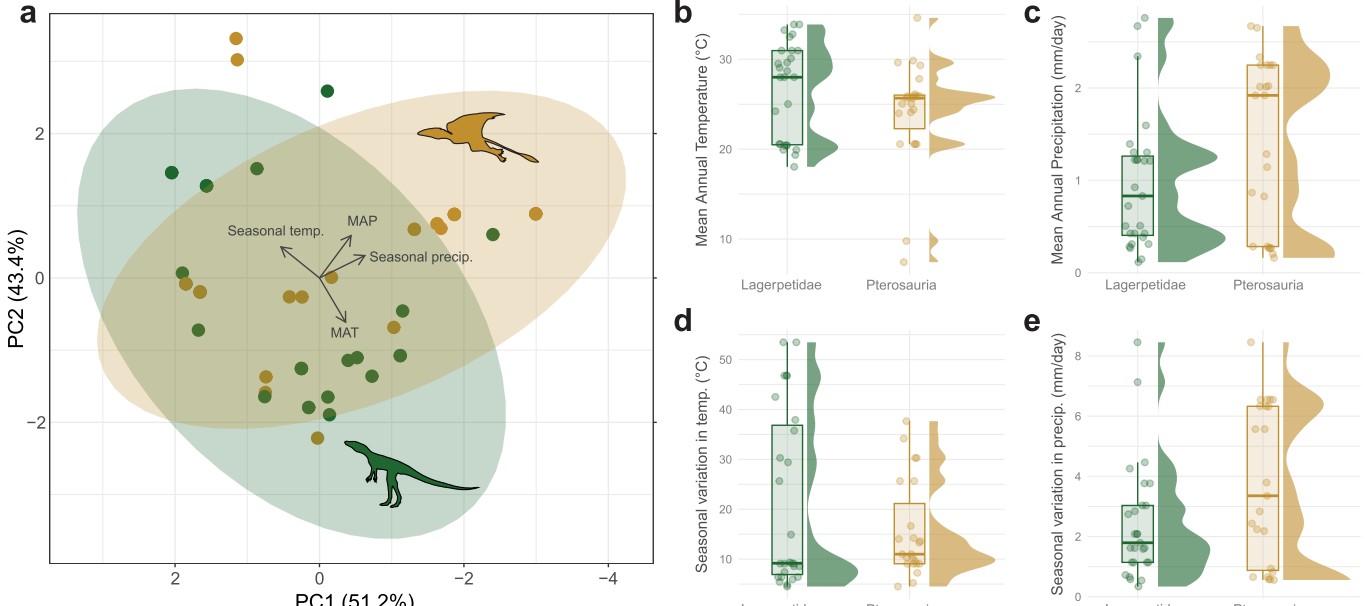

**Extended Data Fig. 2 | Climatic preferences of Middle–Late Triassic pterosauromorph lineages. a**, Principal Component Analysis (PCA) of variables of climatic importance showing early pterosaurs (n = 23) and all lagerpetids (n = 26). **b–e**, raincloud plots comparing climatic variables between early pterosaur and all lagerpetids. Results of statistical comparisons are reported in Supplementary Table 2. Abbreviations: MAP, Mean Annual Precipitation; MAT, Mean Annual Temperature; P, precipitation; T, temperature. The box plots display the median and IQR, the upper whiskers extends from the 75th percentile to the largest value and the lower whiskers extends from the 25th percentile to the lowest value.

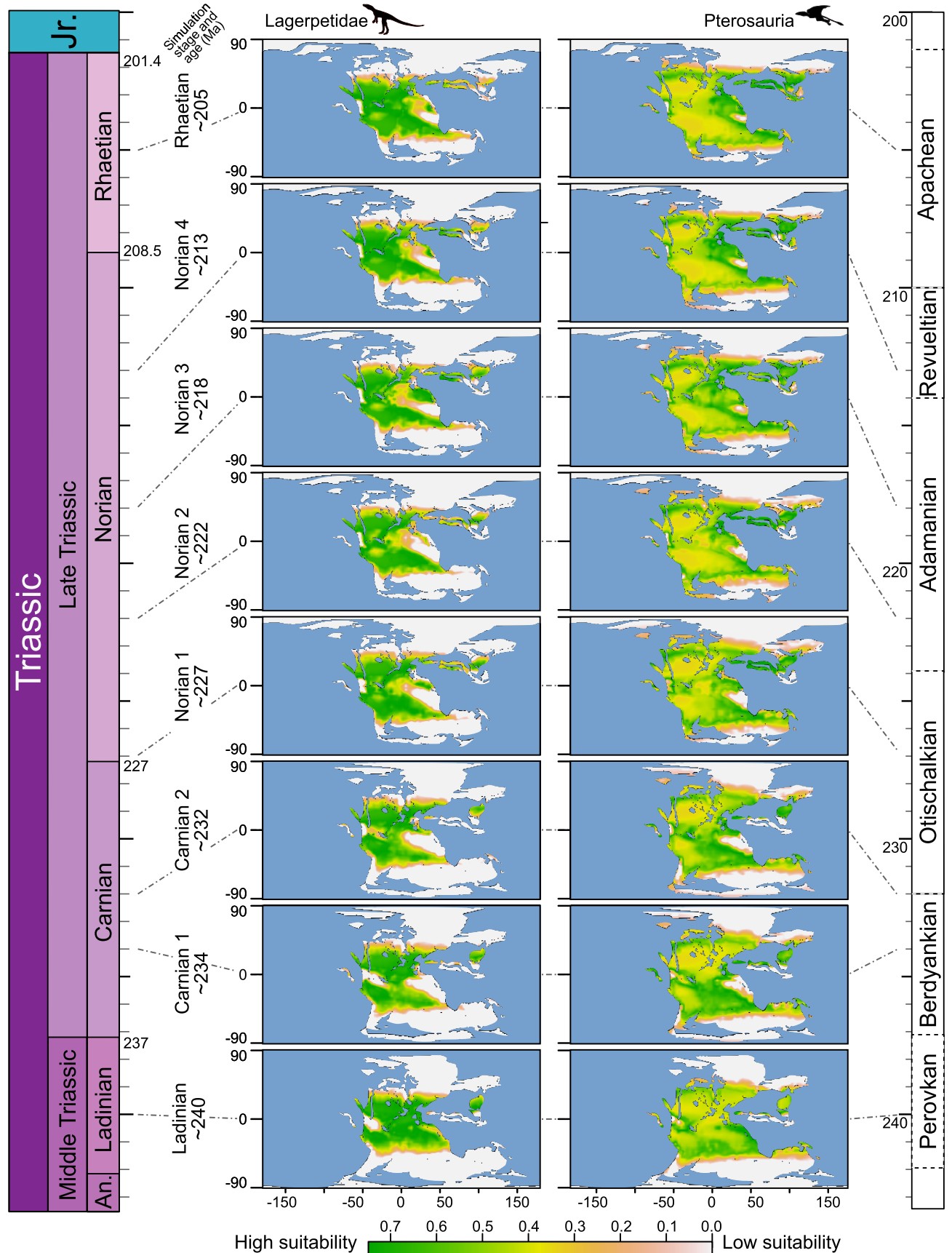

**Extended Data Fig. 3 | Climatic suitability of early pterosaurs and lagerpetids through the Anisian–Rhaetian interval.** The symbol * indicates the reference stage used to generate models for the others. Note that there is, to date, no Ladinian–Carnian fossil record of pterosaurs and no Anisian body fossil of lagerpetids.

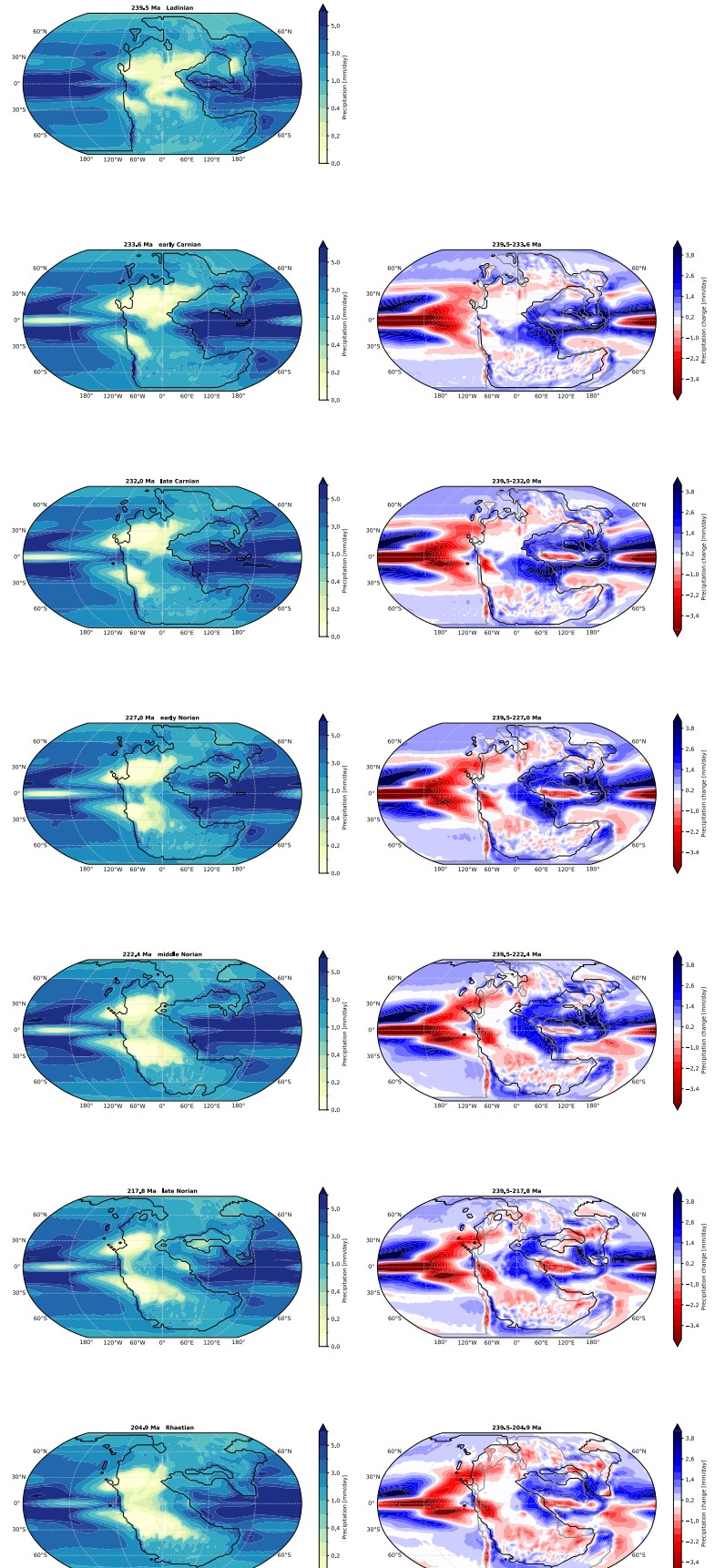

**Extended Data Fig. 4 | Absolute precipitation (left) and anomaly precipitation (right) relative to Ladinian.** Note the increased precipitation (blue areas on the right column plots) along the Tethyan Gulf starting in the Carnian, and increased aridity of the equatorial Pangaea in the middle Norian–Rhaetian.

**Extended Data Table 1 | Statistical comparison using one-sided Wilcoxon rank sum (Mann-Whitney) tests of the climate ranges occupied by: (a) pre-Norian (Ladinian–Carnian; L-C) lagerpetids and lagerpetids from the Norian–Rhaetian (N-R); (b) lagerpetids and pterosaurs, both during the Norian–Rhaetian (N-R); (c) all lagerpetids and pterosaurs**

| Variables ↓ | Lagerpetidae (L-C) vs Lagerpetidae (N-R) | Pterosauria vs Lagerpetidae (N-R) | Pterosauria vs Lagerpetidae (all) |
|---|---|---|---|
| **Mean Annual Temperature [MAT]** | W = 15, **p = 0.0009762*** ** | W = 312.5, **p = 0.001418** ** | W = 375.5, p = 0.2499 |
| **Mean Annual Precipitation [MAP]** | W = 108, p = 0.0938 | W = 146.5, p = 0.184 | W = 238.5, p = 0.229 |
| **Seasonal variation in temperature** | W = 141, **p = 0.0005413*** ** | W = 114.5, **p = 0.02746** ** | W = 298.5, p = 1 |
| **Seasonal variation in precipitation** | W = 69, p = 0.7051 | W = 157.5, p = 0.3044 | W = 227.5 p = 0.1546 |

Asterisks indicate significance levels * → 0.05, ** → 0.01, *** → 0.001.

**Extended Data Table 2 | Results for the Kruskal-Wallis and Dunn's test of the separation of pterosaurs and all lagerpetids for each climate variable**

| Climate variable | Group 1 | Group 2 | n1 | n2 | eff. size | stat | p value | p adj. | p adj. signif. |
|---|---|---|---|---|---|---|---|---|---|
| ➤ **Mean Annual Temperature (°C) [MAT]** | Lagerpetidae | Pterosauria | 24 | 20 | small | -1.20 | 0.229 | 0.229 | ns |
| ➤ **Seasonal difference in temperature (°C)** | Lagerpetidae | Pterosauria | 24 | 20 | small | -0.0236 | 0.981 | 0.981 | ns |
| ➤ **Mean Annual Precipitation (mm/day) [MAP]** | Lagerpetidae | Pterosauria | 24 | 20 | small | 1.84 | 0.657 | 0.657 | ns |
| ➤ **Seasonal difference in precipitation (mm/day)** | Lagerpetidae | Pterosauria | 24 | 20 | moderate | 2.34 | 0.0195 | 0.0195 | * |
| Mean tempearture coldest season (°C) | Lagerpetidae | Pterosauria | 24 | 20 | small | -0.968 | 0.333 | 0.333 | ns |
| Mean tempearture warmest season (°C) | Lagerpetidae | Pterosauria | 24 | 20 | large | -3.80 | 0.000145 | 0.000145 | *** |
| Mean precipitation wet season (mm/day) | Lagerpetidae | Pterosauria | 24 | 20 | moderate | 2.38 | 0.0171 | 0.0171 | * |
| Mean precipitation dry season (mm/day) | Lagerpetidae | Pterosauria | 24 | 20 | moderate | 2.55 | 0.0108 | 0.0108 | * |

Asterisks indicate significance levels * → 0.05, ** → 0.01, *** → 0.001. Abbreviations: eff. size, effect of the difference in sample size; n, number of observations; ns, non-significant results; p.adj, adjusted p values. The symbol ➤ and bold font indicate the climate variables that were used in the palaeoclimate niche occupation multivariate analysis, while those underlined were used to produce the habitat suitability analyses.

# Reporting Summary

## Statistics

For all statistical analyses, confirm that the following items are present in the figure legend, table legend, main text, or Methods section.

| n/a | Confirmed | |
|---|---|---|
| ☐ | ☒ | The exact sample size (*n*) for each experimental group/condition, given as a discrete number and unit of measurement |
| ☒ | ☐ | A statement on whether measurements were taken from distinct samples or whether the same sample was measured repeatedly |
| ☐ | ☒ | The statistical test(s) used AND whether they are one- or two-sided<br>*Only common tests should be described solely by name; describe more complex techniques in the Methods section.* |
| ☐ | ☒ | A description of all covariates tested |
| ☒ | ☐ | A description of any assumptions or corrections, such as tests of normality and adjustment for multiple comparisons |
| ☒ | ☐ | A full description of the statistical parameters including central tendency (e.g. means) or other basic estimates (e.g. regression coefficient) AND variation (e.g. standard deviation) or associated estimates of uncertainty (e.g. confidence intervals) |
| ☐ | ☒ | For null hypothesis testing, the test statistic (e.g. *F*, *t*, *r*) with confidence intervals, effect sizes, degrees of freedom and *P* value noted<br>*Give P values as exact values whenever suitable.* |
| ☒ | ☐ | For Bayesian analysis, information on the choice of priors and Markov chain Monte Carlo settings |
| ☒ | ☐ | For hierarchical and complex designs, identification of the appropriate level for tests and full reporting of outcomes |
| ☒ | ☐ | Estimates of effect sizes (e.g. Cohen's *d*, Pearson's *r*), indicating how they were calculated |

*Our web collection on statistics for biologists contains articles on many of the points above.*

## Software and code

Policy information about availability of computer code

| | |
|---|---|
| Data collection | The first iteration of occurrence data was done using the Palaeobiology Database. This first search proved inadequate for the scope of the study and was updated and expanded through literature search by DF.<br>No other software was used to collect data. |
| Data analysis | All analyses were performed using freely available software, and packages in R / R studio. Custom codes and data will be made available in a specified repository (figshare), alongisde pertinent data) |

For manuscripts utilizing custom algorithms or software that are central to the research but not yet described in published literature, software must be made available to editors and reviewers. We strongly encourage code deposition in a community repository (e.g. GitHub). See the Nature Portfolio guidelines for submitting code & software for further information.

## Data

Policy information about availability of data

All manuscripts must include a data availability statement. This statement should provide the following information, where applicable:
- Accession codes, unique identifiers, or web links for publicly available datasets
- A description of any restrictions on data availability
- For clinical datasets or third party data, please ensure that the statement adheres to our policy

Data have been made available to reviewers and will be made available upon accepance in a specified repository (figshare).

# Research involving human participants, their data, or biological material

Policy information about studies with [human participants or human data](). See also policy information about [sex, gender (identity/presentation), and sexual orientation]() and [race, ethnicity and racism]().

| | |
|---|---|
| Reporting on sex and gender | n/a |
| Reporting on race, ethnicity, or other socially relevant groupings | n/a |
| Population characteristics | n/a |
| Recruitment | n/a |
| Ethics oversight | n/a |

Note that full information on the approval of the study protocol must also be provided in the manuscript.

# Field-specific reporting

Please select the one below that is the best fit for your research. If you are not sure, read the appropriate sections before making your selection.

☐ Life sciences      ☐ Behavioural & social sciences      ☒ Ecological, evolutionary & environmental sciences

For a reference copy of the document with all sections, see [nature.com/documents/nr-reporting-summary-flat.pdf]()

# Ecological, evolutionary & environmental sciences study design

All studies must disclose on these points even when the disclosure is negative.

| | |
|---|---|
| Study description | This study is 2-fold:<br>1) Palaeobiogeography. This section utilises phylogenetic relationships of pterosaurs and allies to determine their evolutionary history and palaeobiogeography. Specifically, we assembled a supertree, assigned each tip to a geographical area and calculated the potential of dispersal and extend of latitudinal dispersal of pterosaromorphs, compared to close allies.<br>2) Using a newly assembled dataset of Triassic pterosauromorphs we calculated the climate preference of lagerpetids vs pterosaurs, using climate variables calculated for each locality of occurrence. The climatic preference of each group was first assessed using multivariate analyses, and then projected in palaeogeographic maps to assess the distribution of high-suitability areas through time during the Middle to Late Triassic. |
| Research sample | Unique single occurrence of Triassic pterosauromorphs. |
| Sampling strategy | For each locality at least one specimen, positively identifiable as either a lagerpetid or a pterosaur was included in the manuscript. Where multiple specimens occurred in the same locality, only one occurrence per group was considered.<br>To avoid redundacy, a single locality was considered if two or more localities resulted indistinguishable due to the resolution of the climate analysis grid. This was the case, unless distinct taxa occurred in the cluster; in that case individual unique occurrences were considered. |
| Data collection | Occurrence data of pterosauromorphs were collected through the Paleobiology Database (PBDB). This initial search was expanded with literature search, and new occurrences were obtained through fieldwork data from Petrified Forest National Park, Arizona, USA. DF lead the collection of the occurrence data of Triassic pterosauromorphs.<br>Climate data were produced by AF (see methods), and utilised by AAC and EMD for further analyses. |
| Timing and spatial scale | Our data include pterosauromorphs of the Triassic period. Our study covers the Anisian-Rhaetian stage. |
| Data exclusions | No occurrence data were excluded except in case it was redundant (e.g., two specimens of the same taxon from the same locality corresponded to a single entry). |
| Reproducibility | Data and code are available. |
| Randomization | Datapoints were assigned to one of two groups based on positive morphological criteria (= diagnostic features). |
| Blinding | n/a |

Did the study involve field work?      ☐ Yes      ☒ No

# Reporting for specific materials, systems and methods

We require information from authors about some types of materials, experimental systems and methods used in many studies. Here, indicate whether each material, system or method listed is relevant to your study. If you are not sure if a list item applies to your research, read the appropriate section before selecting a response.

## Materials & experimental systems

| n/a | Involved in the study |
|---|---|
| ☒ | Antibodies |
| ☒ | Eukaryotic cell lines |
| ☐ | ☒ Palaeontology and archaeology |
| ☒ | Animals and other organisms |
| ☒ | Clinical data |
| ☒ | Dual use research of concern |
| ☒ | Plants |

## Methods

| n/a | Involved in the study |
|---|---|
| ☒ | ChIP-seq |
| ☒ | Flow cytometry |
| ☒ | MRI-based neuroimaging |

## Palaeontology and Archaeology

**Specimen provenance**
*Provide provenance information for specimens and describe permits that were obtained for the work (including the name of the issuing authority, the date of issue, and any identifying information). Permits should encompass collection and, where applicable, export.*

**Specimen deposition**
*Indicate where the specimens have been deposited to permit free access by other researchers.*

**Dating methods**
*If new dates are provided, describe how they were obtained (e.g. collection, storage, sample pretreatment and measurement), where they were obtained (i.e. lab name), the calibration program and the protocol for quality assurance OR state that no new dates are provided.*

☒ Tick this box to confirm that the raw and calibrated dates are available in the paper or in Supplementary Information.

**Ethics oversight**
*Identify the organization(s) that approved or provided guidance on the study protocol, OR state that no ethical approval or guidance was required and explain why not.*

Note that full information on the approval of the study protocol must also be provided in the manuscript.

## Plants

**Seed stocks**
n/a

**Novel plant genotypes**
n/a

**Authentication**
n/a

