## [Peer Review File · Nature Ecology & Evolution]

Climate drivers and palaeobiogeography of lagerpetids and early pterosaurs

Corresponding Author: Dr Davide Foffa

Version 0:

Decision Letter:

16th April 2024

Dear Dr Foffa,

Your Article, "Climate drivers and paleobiogeography of lagerpetids and early pterosaurs" has now been seen by three reviewers. You will see from their comments copied below that while they find your work of considerable potential interest, they have raised quite substantial concerns that must be addressed. In light of these comments, we cannot accept the manuscript for publication, but would be very interested in considering a revised version that addresses these serious concerns.

We hope you will find the reviewers' comments useful as you decide how to proceed. If you wish to submit a substantially revised manuscript, please bear in mind that we will be reluctant to approach the reviewers again in the absence of major revisions.

In particular, the reviewers raise concerns about methods reporting (particularly as regards phylogenetic justification), correcting for sampling bias, and the need to compare with more extensive proxy data--on that front, while attending to the various technical recommendations made by all three reviewers will be necessary, for this paper to succeed at Nature Ecology & Evolution, we ask you to pay particular attention to reviewer 1's recommendations to substantiate your interpretations with further proxy use and a more ecological framing.

If you choose to revise your manuscript taking into account all reviewer and editor comments, please highlight all changes in the manuscript text file [OPTIONAL: in Microsoft Word format].

* Include a "Response to reviewers" document detailing, point-by-point, how you addressed each referee comment. If no action was taken to address a point, you must provide a compelling argument. This response will be sent back to the referees along with the revised manuscript.

* If you have not done so already we suggest that you begin to revise your manuscript so that it conforms to our Article format instructions at <http://www.nature.com/natecolevol/info/final-submission>. Refer also to any guidelines provided in this letter.

Link Redacted

If you wish to submit a suitably revised manuscript we would hope to receive it within 6 months. If you cannot send it within this time, please let us know. We will be happy to consider your revision so long as nothing similar has been accepted for publication at Nature Ecology & Evolution or published elsewhere.

Nature Ecology & Evolution is committed to improving transparency in authorship. As part of our efforts in this direction, we are now requesting that all authors identified as 'corresponding author' on published papers create and link their Open Researcher and Contributor Identifier (ORCID) with their account on the Manuscript Tracking System (MTS), prior to acceptance. This applies to primary research papers only. ORCID helps the scientific community achieve unambiguous attribution of all scholarly contributions. You can create and link your ORCID from the home page of the MTS by clicking on 'Modify my Springer Nature account'. For more information please visit www.springernature.com/orcid.

Thank you for the opportunity to review your work.

[redacted]

Reviewer expertise:

Reviewer #1: Palaeozoic and Mesozoic climates and biogeography

Reviewer #2: macroevolutionary methods and phylogenetics

Reviewer #3: pterosaurs

Reviewers' comments:

Reviewer #1 (Remarks to the Author):

To the authors:

I congratulate you on the attempt to link pterosaur dispersal to the dynamics of the Late Triassic climate, but I am a bit disappointed by the results. First, you rely heavily on the results of the HadCM3L-M2 model run to interpret climate, but other than the humid Carnian interval, invoke no proxies from the stratigraphic record. The changes from stage to stage calculated by the model are so subtle that reference to actual formations and the proxies contained therein would make your interpretations more credible.

Secondly, your work would benefit from a more ecological viewpoint. Why do organisms occupy specific habitats? Generally for feeding (and reproductive) purposes. Did pterosaurs evolve more diverse feeding strategies during the 20 My of their Late Triassic history? And would these new strategies allow them to occupy more diverse environments?

More specific comments below:

Abstract

Line 42-43. "We then explored the climatic preferences of each group by integrating fossil occurrences with palaeoclimate"

>>Technically, no. The fossil occurrences are compared to a (very coarse resolution) paleoclimate model. It does not help that some of the figures illustrate fossil occurrence data or environmental suitability compounded for the entire 20 My of the Norian stage.

Line 45. "Lagerpetids, resilient to climatic barriers..."

>>Do you ever specify what these climate barriers might be? Presumably drier climate presented a dispersal barrier to pterosaurs but not lagerpetids.

Lines 47-50. "Conversely, pterosaurs favoured wetter regions, and diversified from low to high latitudes as climatic conditions changed following more favourable conditions (long term increased humidity) in the Late Triassic. Spatial segregation and climatic niche distinctions characterized early pterosaur evolution. Our study suggests that a significant environmental disturbance, likely triggered by alterations of thermal constraints to CO₂ and/or palaeogeography in the early Late Triassic, played a key role in shaping the initial evolution of this iconic vertebrate group."

>>Very vague wording. To what environmental disturbance does this refer? The Carnian Pluvial Episode?

Palaeoclimate occupation

>>This section (lines 153-167) would carry more weight with more references to the specific formations in which the

specimens were found. For example, (Line 160) *Caelestiventus* is from the Nugget Sandstone (a Wingate equivalent) and spans the Rhaetian to Hettangian boundary. It is found associated with *Grallator* tracks. Is the Rhaetian age confirmed?

Ecological niche modeling / Palaeobiogeography, palaeoclimate and ecological niches

Lines 192-196 “During the Norian (our ‘training stage’; Fig. 4c–d), pterosaurs (now represented by body fossils) were widely distributed in coastal, wet, tropical, low-latitude areas, with lower climatic suitability towards the higher latitudes of their ranges and very low suitability in western Pangaea (North and South America). This is accompanied by higher climatic suitability across broader latitudes and a higher predominance in North and South America.”

>>This assumes climate as the only factor involved in the dispersal of the pterosaurs, ignoring a primary factor in their niche partitioning – feeding strategies. Early pterosaurs were unquestionably aerial piscivores, explaining their limitation to wet, coastal environments, while lagerpetids were cursorial and able to occupy a much wider range of environments, including drylands. It is not just possible, but very likely that pterosaur evolutionary adaptations included varied feeding strategies that allowed them to occupy a wider range of environments.

Lines 235-240. “Another possibility is that during this early stage of their evolutionary history, owing to their predominantly arboreal ecomorphology and yet-to-be-fully-attained active flight capabilities¹, their biogeography was constrained by the distribution of the canopy habitat they inhabited. As more suitable areas expanded by the Norian (Fig. 4), pterosaurs may have expanded latitudinally, benefiting from potentially increased suitable habitats.”

Lines 290-292. “This could support the idea that pterosaurs originated shortly before their first appearance in the fossil record and/or that pterosaurs originated earlier but did not have a global distribution until later in the Norian.”

>>The oldest pterosaurs found were already highly derived, so this is very unlikely.

Lines 334-336. “By contrast, early pterosaurs had a more constrained distribution in terms of mean annual and seasonal temperatures, favouring wet areas, which became more common after the Carnian, spreading throughout the latest Triassic.”

>>According to the precipitation models presented, the Rhaetian is the only stage that shows an increase in absolute precipitation over a substantial area compared to the previous stage. There were only slight precipitation increases in early and middle Norian, and these were limited to the Tethyan region. The late Norian shows a general decrease in precipitation over a broad area. Overall, most of Pangaea gets drier during the Norian stage as a whole, as compared to the late Carnian. Thus, the dispersal of the pterosaurs very likely involves adaptations to different environments in addition to (or even instead of) the spread of more suitable climate habitat.

Lines 448-449. “This integrated a literature search with unpublished fieldwork data...”

>>Citations for this literature?

Supplementary Information

Lines 68-75. “Palaeoclimate model simulation uncertainties Databases of global climate proxy data are available for past time periods; however, proxy-evidence becomes increasingly less constrained, and the amount of spatiotemporal data diminishes, increasingly further into the geologic past. This necessitates the use of palaeoclimate model simulations, to provide global coverage. However, palaeoclimate model simulations of deep-time climates are challenging, and themselves have associated uncertainties. These uncertainties can in general be partitioned into two main sources, i) boundary condition uncertainty, and ii) global climate model uncertainty.”

>>The HadCM3L-M2 model is state of the art for this type of simulation, but the results still require evaluation. It is a positive step that the authors make a careful comparison of their model with proxy data from the mid-Carnian Pluvial (Ruffell et al., 2015), and as a suggestion, the GSL published an entire themed volume on the Carnian Pluvial Episode in 2018.

But why stop with the mid-Carnian? We have a wealth of proxy data for North America including the Colorado Plateau (Chinle and Dockum Groups), eastern North America (Newark Supergroup), South America is represented by Argentina (the Ishigualasto and Rio Colorado formations in the Ischigualast Basin), and Europe by the Germanic Basin (Keuper). For many of these, there are extant studies including estimates of MAT and/or MAP. Why not use at least some of these as a check on the model? The record of the Norian in particular is dynamic and complex than can be captured with a model of this resolution. One starting point might be Sun et al. (2020; EPSL). By relying exclusively on the model to interpret palaeoclimate, I am afraid the authors miss seeing a more complete and complex picture.

Lines 118-124. “Greenhouse gas concentrations, more specifically, pCO₂ concentrations, vary through the geologic past, due to long-term imbalances between tectonic emissions and weathering. CO₂ concentrations can be estimated from CO₂ proxies, but proxy type, age, techniques, and calibration uncertainty, as well as temporal sparsity of records, can all make

constraining a deep time pCO₂ concentration problematic. Here we use the CO₂ record of Foster, et al. (2017), which accounts for some of the issues above, but is still associated with substantial uncertainties”

>>This is as good an approach as any. There are MANY pCO₂ reconstructions available, and they all come with uncertainties that depend entirely on which proxies are being used.

Reviewer #2 (Remarks to the Author):

Dear Editor,

This study examines the paleobiogeography of lagerpetids and pterosauroforms in the Triassic. The authors synthesize recent work and make some interesting points regarding the distribution of fossil occurrences. There appear to be multiple deficiencies in the methods, though the presentation makes it difficult to understand exactly how the study was conducted. The presentation isn't convincing enough to warrant publication, and even upon suitable revision, the premise of the paper does not rise to the standard of a Nature journal (see comments below).

Major Comments

The very premise of the paper (from the abstract: “Despite the long temporal overlap between these two clades in the Late Triassic, their limited sympatry implies that ecological partitioning, potentially linked to global palaeoclimate, influenced their early evolution.”) is not remotely interesting. Pterosaurs were flying vertebrates, whereas lagerpetids were small cursorial animals – of course, they occupy different niches. The author’s setup would be akin to asking if mice and bats are partitioned into different niches.

The authors do not demonstrate that their results are not an artifact of sampling and fossil record biases – a known and major confounder for paleogeographical studies. They do not include any sort of sampling bias metric in their models and only briefly acknowledge the possibility that their results are influenced by “severe sampling biases”. Given the paucity of early pterosaurs, sampling within the basal-most portion of the pterosaur tree is going to be sparse, as noted by the authors, which will inherently affect interpretations of biogeography. Similarly, lagerpetids are a relatively small and species-poor clade, perhaps also indicative of fossil record biases. Proxies of fossil record bias should be included in the analyses.

Inadequate quantitative results or statistical tests to back up the claims being made: “ Δ -log likelihood” is discussed in the Palaeobiogeography results section, but no numbers are given – no parameter estimates are given anywhere. Δ -log likelihood is usually a measure of the difference in fit between two models. I think that is the case for Biogeobears (though I have never used it). Moreover, there are no quantitative (statistical tests) in this section or the Results section on Ecological Niche Modelling. Qualitative descriptions are inadequate. Is the occurrence of the two groups really “significant” and what parameter values are the models producing under what assumptions?

Related to the above point, a phylogenetic tree is chosen but doesn’t appear to be used in any of the analyses. This is a major potential flaw.

Specific comments

Abstract

Lines 36-37: “Temporal” would be more appropriate than “stratigraphic” in this context as it is not referring specifically to rock or outcrop, but rather overall disparity in the age of fossils (there are other instances of this as well).

Lines 36-40: It is stated that there are “stratigraphic and morphological gaps” that separate pterosaurs from lagerpetids, then immediately after it is immediately stated that the groups have “long temporal overlap”. These two statements are opposite one another, so it would clarify things immensely to give some specific geological time frames when they do and do not overlap in time and space.

Lines 49-50: “Spatial segregation and climatic niche distinctions characterized early pterosauroform evolution.” This feels very abrupt and is perhaps too definite a statement. I would recommend adding nuance with some sort of lead-in, perhaps, “These findings suggest that spatial segregation...”

Main

Lines 86-89: “Lagerpetids are found in continental fluviolacustrine deposits on at least current four continents and maintained a wide latitudinal and geographic spread throughout their evolutionary history (late Ladinian–late Norian, 237–211 Ma; Fig. 1).” Lagerpetids do not go extinct until the end-Triassic (~201 Ma) and are shown in the Rhaetian map in Figure 1.

Lines 91-92: “From the upper Norian onward, pterosaurs spread to higher latitudes and in a broader array of habitats”. Figure 1 makes it look like pterosaurs are consistently concentrated at mid latitudes. One taxon, *Arcticodactylus*, occurs northward in the Rhaetian, but to say that pterosaurs are spreading northwards based on a single taxon feels like an overgeneralization.

Results

Lines 114-117: "Our model suggests that lagerpetids – and pterosauromorphs as a whole – likely originated in southwestern Pangaea (i.e., modern South America), whereas the origin of pterosaurs was found at low latitudes in the Northern Hemisphere (ED Fig. 1), consistent with previous studies." Of course, this is where the model will place the origin of these groups, as this is where all the fossils are. Some nuance should be added to this statement, as these origins are entirely based on the tree and sampling of fossils.

Lines 159-161: "There are outliers to this pattern, namely *Arcticodactylus* from Greenland, *Caelestiventus* from the southwestern USA, and *Pachagnathus* and *Yelaphomte* from Argentina". Four outliers is quite a large percentage of a dataset with only 29 taxa (~14%). Are they really outliers, or is this just another instance of sampling bias?

Lines 165-167: "It is possible that these high latitude pterosaur occurrences are evidence of the clade's first dispersal event outside of its hypothesised low-latitude ancestral area (see Discussion)." These pterosaur occurrences are at a lower paleolatitude than the "ancestral area" near the Tethys.

Lines 178-180: "Pterosaur potential distribution appears overall patchier and restricted latitudinally, spreading out only from the Carnian, compared to the more homogeneous and latitudinally broader distribution of lagerpetids." According to Figure 1 and the supplementary data, pterosaurs actually have a broader latitudinal range than lagerpetids by about 10 degrees (Lagerpetids: -49.13 to 34.8 paleolat; Pterosaurs: -47.18 to 43.91 paleolat).

Discussion

Lines 230-235: "The preference of pterosaurs for wet, tropical low latitude areas before the Norian may imply a narrower, more specialised climatic preference, and/or that these suitable conditions were initially limited to equatorial areas. Nevertheless, suitable conditions for pterosaurs are found globally in the upper Norian–Rhaetian (Fig. 4b,d), in accordance with the pattern of quantified latitudinal dispersal and increased geographical spread in the fossil record (Fig. 1)." If there are plenty of "suitable conditions" found globally for pterosaurs in the Norian, but they are only found in the Tethys region, is that not very strong evidence for sampling bias?

Lines 266-267: "However, it is worth noting that while there is likely a genuine biogeographical signal in these observations, they could also reflect severe sampling biases". This is the first time sampling bias is mentioned. If it is acknowledged that there could be severe sampling biases, why are they not accounted for in the model?

Methods

Given the statement about there being severe sampling biases, I would like to see more investigation into how such biases impact the results and conclusions reported here. I recommend an approach outlined by Dunne et al. (2021), which standardized fossil collections with geographic space and Gardner et al. 2019, which studied how fossil record biases affect dispersal estimates. Otherwise, a compelling reason to exclude such analyses should be stated.

A more detailed justification for the tree used in the model should be included. The ability of the model to infer ancestral locations and dispersal rates is entirely dependent on the topology of the tree, especially for poorly sampled groups such as early pterosauromorphs. I recognize that no tree will ever be perfect, but I think that a discussion of the various trees that exist and potential points of ambiguity is warranted. As of now, the methods simply state that analyses are based on "...a randomly selected most parsimonious tree of Archosauromorpha...(line 356)". Do the results change when a different most parsimonious tree is selected? How many most parsimonious trees exist? I would argue it would also be good to run an analysis using a strict consensus, removing as much ambiguity as possible, and seeing how that impacts results.

Figures

Figure 1: Despite limited sympatry of pterosaurs and lagerpetids being a central point of this study, Figure 1 makes it look like they almost always co-occur, and no statistics tests are given one way or another. I would recommend modifying the presentation of data, perhaps with zoomed in panels so that it is clear they are not occurring in the same localities. It is also worth considering breaking the maps up into finer temporal resolution. Changes in biogeography within stages are frequently discussed, so a maps for entire stages may be misleading. For instance, with lines 89-91: "Conversely, the earliest body fossils of unambiguous pterosaur are middle Norian in age and found only in a series of marine strata in a low latitude area bordering the Tethys Ocean." I was initially confused as additional pterosaur occurrences are shown in Figure 1b in North America, far away from the Tethys. I understand after reading in more detail that the North American pterosaurs are Late Norian, versus the middle Norian taxa near the Tethys, but to avoid any misinterpretation I think some sort of additional temporal component needs to be added to this figure. Perhaps adding an additional color scale to the points for early, middle, and late Norian (same for Landinian, Carnian, Rhaetian) could add a layer of clarity.

Extended data

Extended Data Figure 1: *Faxinalipterus minimus* is tip dated to ~215 Ma, but the most recent study for this taxon provides a U-Pb date of 225.42 for the locality that produced it (Kellner et al., 2022). Given that it is seemingly a basal taxon, this younger date could be greatly impacting model results. However, even if the date of 225 is more accurate, it is still relatively geologically younger than other lagerpetids. When relatively late-occurring species in a clade are recovered in very basal

positions, it's worth examining the possible causes, and asking if that position is realistic or possibly an artifact of some biases. One explanation might be a genuine ghost lineage. Other causes might include very incomplete specimens which are minimally comparable to other OTUs in terms of character representation. Further, it is very well documented now that immature archosaurs will often be recovered in basal phylogenetic positions due to expression of ancestral or basal features early in growth. The very small size of the holotype of *Faxinalipterus minimus* (giving rise to the species name itself) urges some caution that this may be an immature individual and that its very basal position is an artifact of this immaturity. I know this paper is not intended to re-evaluate every taxon included, but in these biogeography models, tree topology and internal node structure make all the difference when it comes to inferring ancestral node locations and number of dispersal events. If the topology is affected by biases such as ontogeny or specimen incompleteness, the biogeographic results will be affected. I would strongly encourage some nuanced discussion about how biases such as incomplete species and artificial basal position inferred for immature individuals might affect the results of the analyses that depend on tree topology.

Minor edits

Title: I would recommend using the British English spelling of "palaeobiogeography" for consistency. I would further recommend checking for consistency in the use of British English throughout the manuscript as there are several instances of American English (e.g., characterized instead of characterised).

Line 52: "palaeogeography". Typo, should be "palaeogeography"

Line 60: ">60 million years". I would recommend spelling out "more than"

Lines 63-64: "The fossil records of pterosaurs and their kins...". Pluralization. I would recommend "The fossil record of pterosaurs and their kin..."

Line 70: "This work led to the realisation that lagerpetids as the closest relatives..." Should be "This work led to the realisation that lagerpetids are the closest relatives..."

References

Dunne, E.M., Farnsworth, A., Greene, S.E., Lunt, D.J. and Butler, R.J., 2021. Climatic drivers of latitudinal variation in Late Triassic tetrapod diversity. *Palaeontology*, 64(1), pp.101-117.

Gardner, J., K. Surya, and C. L. Organ (2019). Early tetrapodomorph biogeography: controlling for fossil record bias in macroevolutionary analyses. *Comptes Rendus Palevol*. 18 (7): 693-908. doi.org/10.1016/j.crpv.2019.10.008.

Kellner, A.W., Holgado, B., Grillo, O., Pretto, F.A., Kerber, L., Pinheiro, F.L., Soares, M.B., Schultz, C.L., Lopes, R.T., Araújo, O. and Müller, R.T., 2022. Reassessment of *Faxinalipterus minimus*, a purported Triassic pterosaur from southern Brazil with the description of a new taxon. *PeerJ*, 10, p.e13276.

Reviewer #3 (Remarks to the Author):

To investigate early pterosauiromorph distribution in time and space plus environmental data, this work assumes (as modern consensus) that lagerpetids are the closest relatives of pterosaurs and thus, uniting the two groups in the Pterosauiromorpha, with the Lagerpetidae as the non-flighted forms, whereas pterosaurs are the potentially active flyers. The work, therefore, offers an overview of this distribution based on this assumption, which leaves me curious about how these results might have appeared if another group had been considered instead. But within this scope, I find that the authors effectively handle the results of their analyses and arrive at reasonable, well-funded conclusions.

Regarding the arboreal ecomorphology assumption to early pterosaurs, arboreality has been assumed for some taxa and related to the origin of pterosaur flight, but we still lack quantitative ecomorphological analyses to support this hypothesis. Nevertheless, based on the role that arboreality likely played in the origin of the group, this hypothesis to explain the more constrained distribution for pterosaurs against the lagerpetids' broader distribution and ecological versatility seems plausible.

Therefore, overall, it can be said that the work is quite well done, meticulously crafted with hypotheses grounded in robust and well-justified analyses.

As minor comments:

- at Fig. 1, the originally described material of *Faxinalipterus* (not *Faxinalpterus* as plotted in Fig. 1b) was disassociated into a new putative pterosauiromorph, *Maehary bonapartei* by Kellner et al. (2022), which was considered the earliest-diverging member of Pterosauiromorpha. As the figure shows the Triassic pterosauiromorph occurrences, it would be nice to include *Maehary* overlapping *Faxinalipterus* at the same spot (to improve this diversity along time). This reference was mentioned by the authors, but was *Mehary* indeed considered in this study? (if not, shouldn't it?)

-data of climate niche analysis is empty at figshare; link of climatic suitability analysis (in Data availability) is also missing.

-at the table: there is a double PULR 06 (number) referred to *Lagerpeton chanarensis* to be checked (as mentioned by the

authors); was this checked? It seems to have been something that the authors missed.

Version 1:

Decision Letter:

25th April 2025

Dear Davide

Thank you for submitting your revised manuscript "Climate drivers and palaeobiogeography of lagerpetids and early pterosaurs" (NATECOLEVOL-24020411A). It has now been seen again by the original reviewers (with the exception of reviewer 2, who could not re-review) and their comments are below. The reviewers find that the paper has improved in revision, and therefore we'll be happy in principle to publish it in Nature Ecology & Evolution, pending minor revisions to satisfy the reviewers' final requests and to comply with our editorial and formatting guidelines.

If you have not done so already, please ensure that you also email us completed copies of the Reporting summary and Editorial policy checklists:

Reporting summary: https://www.nature.com/documents/nr-reporting-summary.pdf

Editorial policy checklist: https://www.nature.com/documents/nr-editorial-policy-checklist.pdf

[redacted]

Reviewer #1 (Remarks to the Author):

I congratulate the authors for both the careful responses to the comments of the reviewers (including mine) and the extensive revisions to the manuscript. I am particularly impressed with the new text added to both the main text portion of the manuscript and the supplementary information regarding the palaeoclimate, the computer modeling and the geologic evidence for the palaeoclimate.

A few points I would like to your attention:

RESONSE LETTER

You stated in your letter "we lack the data to investigate whether dietary/habitat/reproductive/physiological aspects were contributing factors." This would certainly be an interesting line for future investigation.

MAIN TEXT

There seems to be some confusion as to the age of the earliest pterosaur fossils. In lines 238-240, the oldest know pterosaur fossils are described as middle Norian in age, including both the northern Tethyan region and the Chinle-Dockum basins. The problem is that in lines 538-542 you cite pterosaur finds from the Tecovas Fm (Dockum) and Blue Mesa Mbr (Chinle), both of which have possible early Norian ages. Please clarify.

SUPPLEMETARY INFORMATION

Same issue appears here in lines 32-34 where you describe the earliest pterosaurs as middle Norian and exclusively from the Tethyan realm. What about the Chinle-Dockum fossils?

lines 170-173. I agree absolutely with this approach. The relationship between orbital cyclicity and climate is complex and the cycle periods are not constant through geologic time (as commonly supposed). The effects of the cycles on insolation are typically small and VERY latitude dependent. Given cycle periods of 10s ky to 100s ky, time averaging insolation is the only practical solution to modeling climate for specific times.

lines 181-183. Foster et al. (2017) is not a bad source for palaeo-pCO₂. The best thing about this study is that it presents the VERY diverging results from the stomatal and soil carbonate proxies. I could point out that there are more recent (although not necessarily better) models of Triassic pCO₂, but it would not be reasonable to ask you to rerun the m model just to use slightly different input.

Reviewer #1 (Remarks on code availability):

I am sorry to say that examination and evaluation of the code used for this research is beyond my area of expertise.

Reviewer #3 (Remarks to the Author):

I am satisfied with the arguments presented and commend the authors for their effort in providing well-supported justifications in response to the issues raised during the review process. I believe the manuscript has the potential to significantly contribute to the ongoing debate regarding the palaeogeography of lagerpetids and pterosaurs.

Reviewer #3 (Remarks on code availability):

At this point, I was able to open all available files with the main manuscript.

Dear Editor and reviewers,

We thank you and the reviewers for the thorough comments on our initial submission. We are pleased to respond to the reviews. We reviewed our paper accordingly and, to the best of our knowledge, have addressed the comments unless otherwise stated. In particular, we have addressed: the reviewers's concerns regarding the stratigraphic resolution of our analyses (with the adoption of an updated version of the general circulation model); the validation of the general circulation (palaeoclimate) model HadCm3L (with extended justifications in the main and supplementary text accompanied with a new figure); provided justification for our choice of phylogenetic dataset for the historical biogeography analyses; and discussed the limitations of the Late Triassic pterosauroform fossil record. All of these modifications resulted in substantial rewriting of the main text, the addition of new supplementary information, and updates to the figures. In the rare instances when we disagree with the suggested modifications, we provide detailed justifications, below.

We thank you for the opportunity to resubmit this work and we hope that, like us, you will consider this new version a substantial improvement of the original manuscript.

Modifications include:

Main text:

- Substantial rewriting/rephrasing of the abstract, discussion, methods, and Extended Data files (hereafter, ED).

Analyses:

- We ran again the analyses of the climate preference and climate-niche analyses using improved versions of the palaeoclimatic model that allowed for a more precise (stratigraphy-wise) extraction of data. The new model simulations have been shown to produce a climate with greater polar amplification, more in line with proxy temperature estimates of warm past periods of Earth's history (see Discussion in the main text and Supplementary information). The resulting ecological modelling results remained largely unchanged.
- Based on the updated model and pterosauroform occurrence list, we revised our multivariate climate occupation. Also in this case, the results remain largely unchanged: the climate space occupancy of lagerpetids and pterosaurs are statistically distinct through their stratigraphic overlap (but their climate space occupation is statistically indistinguishable if the whole evolutionary history of lagerpetids is considered). The main drivers of the differences are temperature and seasonal temperature: in the Norian and Rhaetian, lagerpetids occupied a broader range of temperatures and tendentially warmer regions. These new analyses reveal a change in climate occupation by lagerpetids that in the Ladinian-Carnian were more commonly found in colder areas with broader excursions of seasonal temperature, as opposed to Norian-Rhaetian lagerpetids that are prevalent in warmer and less seasonal areas.
- We also re-ran the biogeographic analyses (correction of *Faxinalipterus minimus* stratigraphic age, modified from ~215 Ma to 225.42 Ma [Kellner et al., 2022]). Results are unaffected, and patterns differ only slightly from the previous analyses. Note that the climate suitability and niche analyses remained unaffected by this issue because both dates for *Faxinalipterus* are within the Norian stage.

Figures:

- Figure 1: We added panels to improve the stratigraphic and geographic visualisation of data following the suggestions of reviewers 1 and 2. Besides improving the visualisation of data, we feel that Figure 1 is a better match with the result and discussion. In detail: we unified the Ladinian and Carnian in Figure 1 and split the Norian into lower, middle and upper (lower Norian = Norian [Otischalkian part], 227–224; 224–215.5 Ma [Adamanian part of the Norian]; upper Norian = Revueltian part of the Norian, 215.5, 208.5 Ma). The patterns of pterosaur distribution are now clearer, with the first occurrences in the middle Norian. To further improve the visualisation, close-up panels were also added to the distribution of pterosauroforms in the American Southwest. This was done following the reviewers' comments to clarify the overlapping clusters in the area in the Norian panel of the first submission. These panels are organised by Land Vertebrate Fauna Chrons, and their approximate correspondence with absolute age and stratigraphic chart is shown.
- Figure 2: We made small modifications to the curves in all three panels as a consequence of the revised age of *Faxinalipterus* (modified from ~215 Ma to 225.42 Ma [Kellner et al., 2022]). These alterations are minor and do not affect the patterns of each group or the overall results of these analyses.
- Figure 3: We remade the figure based on the new climate analyses we ran. We repeated the analyses. We highlighted the climate occupation of Ladinian-Carnian lagerpetids, Norian-Rhaetian lagerpetids (= those that overlap stratigraphically with known pterosaur specimens) and pterosaurs. We also added the climate space for each stage to visualise the changes in climate-space occupation in time.
- Figure 4: We remade the figure based on the new analyses we ran: it now includes more simulation stages, with a unified Ladinian/Carnian Norian is split in three for consistency with Figure 1. All eight stages of the simulations for lagerpetids and pterosaurs are available in the extended version of this figure: ED Fig. 2).

Extended Data

- ED Figure 1: We updated this based on the new age of *Faxinalipterus*.
- ED Figure 2: We created this new figure to show the climate preference of lagerpetids (all) vs pterosaurs.
- ED Figure 3: We remade this figure with the updated general circulation model simulations.
- ED Figure 4: We remade this figure showing absolute precipitation and anomaly precipitation relative to the Ladinian.
- We updated ED Table 1–2, modified ED Table 3 to show the results of the climate niche occupation linked to Figure 3 and added ED Table 4 showing all climate variables.

Supplementary Information

- We modified and expanded the section on the evaluation of the general circulation model simulations and greatly expanded the section discussing the biases in the fossil record of pterosauroforms and how they may affect our analyses (including an additional graph Figure S3).
- Figure S1: We created a new figure exploring the latitudinal occurrences of Middle to Late Triassic tetrapod-, pterosaur- and lagerpetid-bearing localities.
- Figure S2: We remade and expanded the figure to include coal formation predictions and observations from all stages of the Late Triassic.
- We remade Figure S3.

Data and codes

Updated datasets can be found alongside their respective R codes at:

- Potential for latitudinal dispersion: <https://figshare.com/s/7ac05f57d35b3810a2c7>
- Accumulated latitudinal dispersion: <https://figshare.com/s/8cabc46601eb64f9b9f0>
- Palaeoclimate niche occupation: <https://figshare.com/s/6254a692186e79fa9060>
- Habitat suitability: <https://figshare.com/s/6600233d24db8f1d986e>

Reviewer #1: Palaeozoic and Mesozoic climates and biogeography (Remarks to the Author):

To the authors:

I congratulate you on the attempt to link pterosaur dispersal to the dynamics of the Late Triassic climate, but I am a bit disappointed by the results. First, you rely heavily on the results of the HadCM3L-M2 model run to interpret climate, but other than the humid Carnian interval, invoke no proxies from the stratigraphic record. The changes from stage to stage calculated by the model are so subtle that reference to actual formations and the proxies contained therein would make your interpretations more credible.

A. We addressed these concerns in two ways:

1) We have re-run the analyses with an updated version of HadCM3L-M2.1aD. This new version of the model and the associated simulations (see “Scotese07” series of Judd et al, Science, 2024) has greater polar amplification than the previous version, in better agreement with proxy estimates from warm periods such as the early Eocene. This improvement comes largely from tuning some of the model parameterisations associated with clouds. See full discussion in Ross (2022), and in the revised manuscript main paper and Supplementary Information. The resulting ecological niche modelling results are not greatly impacted by the use of the new model simulations.

- Judd, E.J., Tierney, J.E., Lunt, D.J., Montañez, I.P., Huber, B.T., Wing, S.L. and Valdes, P.J. (2024). A 485-million-year history of Earth’s surface temperature. *Science*, 385(6715), p.eadk3705.
- Ross, P. (2022). Deep ocean circulation during the early Eocene: a model-data comparison, PhD thesis, Imperial College, London.
<https://spiral.imperial.ac.uk/entities/publication/694a0346-7cea-4acd-afec-31fb330c8f84>

2) As suggested, we expanded the model-data comparison of coal deposits to all model simulations in the Upper Triassic. Again, we find that for all simulations there is reasonable agreement between the model-predicted coal formation regions and the lithological indicators. In addition, we now draw on previous model-data comparison carried out with the simulations. In particular, Xie et al. (2024) carry out a point-point comparison of the model-predicted evaporites with sedimentological proxies. This can be considered a test of the hydrological cycle in the model. It is not for the Upper Triassic specifically, but for four snapshots in the Phanerozoic, one of which is at 252 Ma. The paper concludes that there is "reasonable agreement" with the proxies. We added a reference to this paper into the manuscript.

- Xie, Y., Lunt, D. J., and Valdes, P. J. (2024) Diagnosing the controls on desert dust emissions through the Phanerozoic, *Clim. Past*, 20, 25612585,
<https://doi.org/10.5194/cp-20-2561-2024>.

To further clarify, the model simulations themselves are not directly tuned to proxy estimates of climate. The model is a numerical representation of the known fluid mechanics and physics of the atmosphere and ocean, all of which is identical to that of modern. The model itself is about 1 million lines of computer code, encapsulating all this theory. In order to simulate a paleo time period, the continental configuration is modified accordingly (e.g.

global maps of mountain position and heights and ocean depths), and the atmospheric concentration of greenhouse gases, and the strength of the sun that the model sees. The model then simulates about 10,000 years of the "weather" of this paleo world (which takes about 12 months of real-time), which are then averaged to produce the climate reconstructions that are finally used. Specifically, the model produces 100's of meteorological variables, of which we only used a handful (e.g. temperature/precipitation). At no point does the model "see" any paleo proxy estimates of temperature and precipitation, but it predicts them based on known physics and fluid mechanics. When the model has finished running, we then compare its output with the geological proxy indicators.

Direct comparisons between geochemical proxies and general circulation models (GCMs) are rare for deep time studies, but some exist. These show that the estimation of palaeoclimate variables extracted from sedimentological, geochemical, and biological proxies are consistent with GCM predictions at basin level. For example, Mancuso and colleagues (2021, 2022) demonstrated that multiproxy from clay mineral geochemical data estimates for mean annual temperature precipitation and high-temperature seasonality from the Chañares-Los Rastros-Ischigualasto succession (early Carnian to early Norian) match the predictions of the GCM used by Dunne et al. (2021), which is a predecessor of the model simulations that we used here. These observations are still valid with the current version of the GCM.

The palaeoclimate of this basin fluctuated from warmer, drier conditions to more temperate humid conditions in the early Carnian to early Norian interval, as broadly predicted by our GCM (see ED Fig. 4). Similarly, Lepre and Olson (2020) used sedimentological and geochemical data to demonstrate a trend of increasing aridity and lower precipitation in the Colorado Plateau Chinle Formation succession (early Norian to Rhaetian). The same trend was recovered by our GCM (see ED Fig. 4). Climate prediction from GCM also corresponds well with inferred Triassic floral provinces (Kustatscher et al., 2018; Mancuso et al. 2021).

- Mancuso, A.C., Irmis, R.B., Pedernera, T.E., Gaetano, L.C., Benavente, C.A., and Breeden III, B.T. (2022). Paleoenvironmental and biotic changes in the Late Triassic of Argentina: testing hypotheses of abiotic forcing at the basin scale. *Frontiers in Earth Science*, 10, 883788.
- Mancuso, A.C., Horn, B.L.D., Benavente, C.A., Schultz, C.L., and Irmis, R.B. (2021). The paleoclimatic context for South American Triassic vertebrate evolution. *Journal of South American Earth Sciences*, 110, 103321.
- Dunne, E.M., Farnsworth, A., Greene, S.E., Lunt, D.J., and Butler, R.J. (2021). Climatic Drivers of Latitudinal Variation in Late Triassic Tetrapod Diversity. *Palaeontology* 64, 101–117. doi:10.1111/pala.12514
- Lepre, C.J., and Olsen, P.E. (2021). Hematite reconstruction of Late Triassic hydroclimate over the Colorado Plateau. *Proceedings of the National Academy of Sciences*, 118(7), e2004343118.
- Kustatscher, E., Ash, S.R., Karasev, E., Pott, C., Vajda, V., Yu, J., and McLoughlin, S. (2018). *Flora of the late Triassic*. The Late Triassic World: earth in a time of transition, 545-622.

Secondly, your work would benefit from a more ecological viewpoint. Why do organisms occupy specific habitats? Generally for feeding (and reproductive) purposes. Did pterosaurs

evolve more diverse feeding strategies during the 20 My of their Late Triassic history? And would these new strategies allow them to occupy more diverse environments?

A. Unfortunately, testing the feeding and reproductive biology of these taxa is currently impossible because of a lack of available data. What we can say is that lagerpetid diets likely consisted of small vertebrates/invertebrates (based on their small body sizes and tooth shape), but only a handful of skulls/jaws are known. Early pterosaur diets are more debated but also likely consisted of small vertebrates/invertebrates and (for some) fish (see Bestwick et al. 2020, Nat. Commun.). Due to the lack of sufficient data/proxies, the changes in diets and their correspondence with geographical distribution would largely remain a qualitative discussion, which we tried to avoid in favour of quantitative analyses based on more certain data (stratigraphic/geographic presence/absence).

- Bestwick, J., Unwin, D.M., Butler, R.J. et al. (2020) Dietary diversity and evolution of the earliest flying vertebrates revealed by dental microwear texture analysis. *Nature Communications* 11, 5293. <https://doi.org/10.1038/s41467-020-19022-2>

We recognise that (syn-)ecology is an important consideration, so we reinforced in the text that the patterns that we see may well be caused by ecological factors, and that these could contribute to explaining the lack of overlap between pterosaurs and lagerpetids.

Regarding habitat preferences (another potential ecological factor): we acknowledged that some (but not all) Triassic pterosaurs are found in coastal environments (matching with our climate suitability prediction), and that, to date, no lagerpetid has. Even if lagerpetids inhabited coastal areas, it'd be more unlikely to find strictly terrestrial animals in the marine deposits where contemporaneous pterosaurs are found. However, this argument is not a good explainer for early pterosauro-morph distribution for three reasons:

- 1) The European (mainly Alpine) geological units in Italy, Switzerland, Austria that preserve pterosaurs have yielded multiple representatives of other unambiguously terrestrial groups (e.g. sphenodontians, drepanosaurs, aetosaurs). This shows that marine habitat does not preclude the preservation of minute, fragile, terrestrial fossils.
- 2) The question “why don't we find pterosaurs in lagerpetid-inhabited areas, during the time overlap of these groups in the Late Triassic?”, cannot be easily dismissed as preservation-bias: the absence of pterosaurs from lagerpetid-bearing formations is unlikely due to preservational biases. Both groups have small and fragile bones, and biases cut both ways. Invoking biases to explain the absence of lagerpetids in pterosaur-bearing formations and ignoring the contrary observation (i.e., the absence of pterosaurs in lagerpetid-bearing formations) is inconsistent.
- 3) The sampling of Triassic pterosauro-morph-bearing geological units is not as bad as it may seem: at least 20 pterosaur specimens are now known from at least 13 Triassic geological units (including continental ones), and 16 unique localities, lagerpetids are known from 16 distinct geological units and 25 unique localities. These are a fraction of the total of extensively sampled Triassic localities and tetrapod-bearing formations. These figures are an underestimate because our dataset ignores occurrences that would be considered redundant in our analyses (e.g., localities that are geographically closer than the cell-resolution of the climate models are counted as one; multiple occurrences of the same taxon in the same locality are counted as one).

None of these arguments is on their own sufficient to justify the patterns that we detect from raw data, but together, they build a picture that we thought worth exploring in this paper.

More specific comments below:

Abstract

Line 42-43. “We then explored the climatic preferences of each group by integrating fossil occurrences with palaeoclimate”

>>Technically, no. The fossil occurrences are compared to a (very coarse resolution) paleoclimate model. It does not help that some of the figures illustrate fossil occurrence data or environmental suitability compounded for the entire 20 My of the Norian stage.

A. We agree that the resolution of the palaeoclimate data is coarse. This is a limitation of the palaeoclimatic reconstructions from the general circulation model. However, we mitigated this by extracting data from updated versions of the general circulation model that have a finer stratigraphic resolution than those reported in the first version of this manuscript. We also added additional fossil occurrences. We modified Figures 1, 2, and 4 to more precisely show stratigraphic and geographic occurrences in the Norian.

Line 45. “Lagerpetids, resilient to climatic barriers...”

>>Do you ever specify what these climate barriers might be? Presumably drier climate presented a dispersal barrier to pterosaurs but not lagerpetids.

A. Yes, that’s our interpretation. Warm and arid climate belts would be the barriers. We clarified this in the abstract and the rest of the text.

Lines 47-50. “Conversely, pterosaurs favoured wetter regions, and diversified from low to high latitudes as climatic conditions changed following more favourable conditions (long term increased humidity) in the Late Triassic. Spatial segregation and climatic niche distinctions characterized early pterosauiromorph evolution. Our study suggests that a significant environmental disturbance, likely triggered by alterations of thermal constraints to CO₂ and/or palaeogeography in the early Late Triassic, played a key role in shaping the initial evolution of this iconic vertebrate group.”

>>Very vague wording. To what environmental disturbance does this refer? The Carnian Pluvial Episode?

A. Yes, the Carnian Pluvial Event would be a strong candidate for the environmental disturbances we referred to in the text. We added explicit references to the CPE in the abstract and the main text. However, at this stage we are reluctant to establish a direct casualty link between the CPE and these events, particularly as this was not the analytical focus of our study.

Palaeoclimate occupation

>>This section (lines 153-167) would carry more weight with more references to the specific formations in which the specimens were found. For example, (Line 160) *Caelestiventus* is from the Nugget Sandstone (a Wingate equivalent) and spans the Rhaetian to Hettangian boundary. It is found associated with *Grallator* tracks. Is the Rhaetian age confirmed?

A. We modified this section, which was incorporated with the first paragraph of the Discussion. As requested, we provided additional references to specific formations of these taxa. *Caelestiventus* was found in Saints & Sinners Quarry, which is either upper Norian or Rhaetian based on the position of the quarry (Triassic–Jurassic boundary within the Nugget Sandstone lies above the quarry) and the presence of other typical groups in the quarry, namely drepanosaurs and procolophonids (Britt et al. 2016).

- Britt et al. (2016) D. Rise of the Erg: Paleontology and Paleoenvironments of the Triassic–Jurassic Transition in Northeastern Utah (Utah Geological Association, Salt Lake City, 2016).

Ecological niche modeling / Palaeobiogeography, palaeoclimate and ecological niches

Lines 192-196 “During the Norian (our ‘training stage’; Fig. 4c–d), pterosaurs (now represented by body fossils) were widely distributed in coastal, wet, tropical, low-latitude areas, with lower climatic suitability towards the higher latitudes of their ranges and very low suitability in western Pangaea (North and South America). This is accompanied by higher climatic suitability across broader latitudes and a higher predominance in North and South America.”

>>This assumes climate as the only factor involved in the dispersal of the pterosaurs, ignoring a primary factor in their niche partitioning – feeding strategies. Early pterosaurs were unquestionably aerial piscivores, explaining their limitation to wet, coastal environments, while lagerpetids were cursorial and able to occupy a much wider range of environments, including drylands. It is not just possible, but very likely that pterosaur evolutionary adaptations included varied feeding strategies that allowed them to occupy a wider range of environments.

A. We rephrased this sentence. We do not intend to imply “[...] *climate as the only factor involved in the dispersal of the pterosaurs*”. This section describes the results of analyses that explicitly test the influence of climate on pterosauro-morph distribution, but we are not excluding other factors.

We briefly touched on this issue in the main text, but we added more detail about diets following this recommendation. Early pterosaurs (including two classically referred to as “piscivores”) fed on a wider variety of prey than fish: Bestwick et al. (2020, Nat. comm.) used dental microwear texture analysis to show that the diet of early pterosaurs was varied (ranging from “soft” invertebrates to vertebrates to hard prey items). Indeed, piscivory (and carnivory) are suggested to be derived from ancestral invertebrate diet. It follows that the pterosaur diet/feeding strategy does not explain their prevalence along coastal areas. Not all early pterosaurs were “coastal”.

We added more nuance to the text regarding other factors that could contribute to the distribution patterns of the two groups. At the moment, however, occurrence data are the only ones we can use to build testable hypotheses. With very few exceptions, we do not find pterosaurs in the same areas where lagerpetids are found and vice versa. This is a striking pattern that is worth exploring, and while we agree that this could be due to a variety of reasons, not just climate, we lack the data to investigate whether dietary/habitat/reproductive/physiological aspects were contributing factors.

Lines 235-240. “Another possibility is that during this early stage of their evolutionary history, owing to their predominantly arboreal ecomorphology and yet-to-be-fully-attained active flight capabilities¹, their biogeography was constrained by the distribution of the canopy habitat they inhabited. As more suitable areas expanded by the Norian (Fig. 4), pterosaurs may have expanded latitudinally, benefiting from potentially increased suitable habitats.”

Lines 290-292. “This could support the idea that pterosaurs originated shortly before their first appearance in the fossil record and/or that pterosaurs originated earlier but did not have a global distribution until later in the Norian.”

>>The oldest pterosaurs found were already highly derived, so this is very unlikely.

A. Our original argument was imprecise. The distribution of canopy habitats becomes more widespread through the Norian with increased wet conditions, which may have favoured an increased distribution of pterosaurs (as a group likely linked to forested environments) through that stage. If suitable canopies were geographically restricted before the Norian, this would have also constrained pterosaur geographical distribution, regardless of their flying capabilities, but especially if early pterosaur flying capabilities were limited.

We rephrased that sentence: *“Another possibility is that during this early stage of pterosaur evolutionary history, perhaps owing to initially constrained flight capabilities, at least partially, confined to arboreal environments¹, their biogeography was constrained by the distribution of the canopy habitat they inhabited.”*

Lines 334-336. “By contrast, early pterosaurs had a more constrained distribution in terms of mean annual and seasonal temperatures, favouring wet areas, which became more common after the Carnian, spreading throughout the latest Triassic.”

>>According to the precipitation models presented, the Rhaetian is the only stage that shows an increase in absolute precipitation over a substantial area compared to the previous stage. There were only slight precipitation increases in early and middle Norian, and these were limited to the Tethyan region. The late Norian shows a general decrease in precipitation over a broad area. Overall, most of Pangaea gets drier during the Norian stage as a whole, as compared to the late Carnian. Thus, the dispersal of the pterosaurs very likely involves adaptations to different environments in addition to (or even instead of) the spread of more suitable climate habitat.

A. This study considers localised occurrences, so average increases over broad areas (continental scale) bear little relevance to this work. A localised increase of suitable habitat conditions in the early and middle Norian would still have allowed the expansion of the geographic range of pterosaurs even if Pangaea was becoming drier overall. This is explained in the text: Lines 267-269 *“However, their climatic preferences might have resulted in a sparse and locally restricted distribution, a possibility confirmed by their modeled, patchily distributed suitable areas.”*

Alternatively, as the reviewer pointed out, the dispersal of pterosaurs could have involved adaptations to different environments; this is something that our data and methods cannot test. However, the distribution of taxa in the PCA data (Fig. 3) does not seem to suggest that's the case: we do not (visually) record a radical change or expansion of the position of Rhaetian pterosaurs compared to Norian ones (Fig. 3f).

Lines 448-449. “This integrated a literature search with unpublished fieldwork data...”

>>Citations for this literature?

A. We added references to the supplementary information files and in this section.

Supplementary Information

Lines 68-75. “Palaeoclimate model simulation uncertainties Databases of global climate proxy data are available for past time periods; however, proxy-evidence becomes increasingly less constrained, and the amount of spatiotemporal data diminishes, increasingly further into the geologic past. This necessitates the use of palaeoclimate model simulations, to provide global coverage. However, palaeoclimate model simulations of deep-time climates are challenging, and themselves have associated uncertainties. These uncertainties can in general be partitioned into two main sources, i) boundary condition uncertainty, and ii) global climate model uncertainty.”

>>The HadCM3L-M2 model is state of the art for this type of simulation, but the results still require evaluation. It is a positive step that the authors make a careful comparison of their model with proxy data from the mid-Carnian Pluvial (Ruffell et al., 2015), and as a suggestion, the GSL published an entire themed volume on the Carnian Pluvial Episode in 2018. But why stop with the mid-Carnian? We have a wealth of proxy data for North America including the Colorado Plateau (Chinle and Dockum Groups), eastern North America (Newark Supergroup), South America is represented by Argentina (the Ishigualasto and Rio Colorado formations in the Ischigualast Basin), and Europe by the Germanic Basin (Keuper). For many of these, there are extant studies including estimates of MAT and/or MAP. Why not use at least some of these as a check on the model? The record of the Norian in particular is dynamic and complex than can be captured with a model of this resolution. One starting point might be Sun et al. (2020; EPSL). By relying exclusively on the model to interpret palaeoclimate, I am afraid the authors miss seeing a more complete and complex picture.

A. See the reply to the first comment. We have expanded our model-data comparison to all stages of the Upper Triassic and include references to previous model-data comparisons in the main text and Supplementary Information.

Lines 118-124. "Greenhouse gas concentrations, more specifically, pCO₂ concentrations, vary through the geologic past, due to long-term imbalances between tectonic emissions and weathering. CO₂ concentrations can be estimated from CO₂ proxies, but proxy type, age, techniques, and calibration uncertainty, as well as temporal sparsity of records, can all make constraining a deep time pCO₂ concentration problematic. Here we use the CO₂ record of Foster, et al. (2017), which accounts for some of the issues above, but is still associated with substantial uncertainties"

>>This is as good an approach as any. There are MANY pCO₂ reconstructions available, and they all come with uncertainties that depend entirely on which proxies are being used.

A: We agree that there is considerable uncertainty associated with the CO₂ reconstruction, and we discuss this in the Supplementary Information. Given the considerable computational cost of carrying out additional general circulation model simulations, it is unfortunately outside the scope of this manuscript to explore this further.

Reviewer #2: macroevolutionary methods and phylogenetics (Remarks to the Author):

Dear Editor,

This study examines the paleobiogeography of lagerpetids and pterosauroforms in the Triassic. The authors synthesize recent work and make some interesting points regarding the distribution of fossil occurrences. There appear to be multiple deficiencies in the methods, though the presentation makes it difficult to understand exactly how the study was conducted. The presentation isn't convincing enough to warrant publication, and even upon suitable revision, the premise of the paper does not rise to the standard of a Nature journal (see comments below).

Major Comments

The very premise of the paper (from the abstract: "Despite the long temporal overlap between these two clades in the Late Triassic, their limited sympatry implies that ecological partitioning, potentially linked to global palaeoclimate, influenced their early evolution.") is not remotely interesting. Pterosaurs were flying vertebrates, whereas lagerpetids were small cursorial animals – of course, they occupy different niches. The author's setup would be akin to asking if mice and bats are partitioned into different niches.

A. This work tries to explain the early distribution through time and space of lagerpetids and pterosaurs, not whether these groups have different ecological niches. Whether pterosaurs were flying and lagerpetids were terrestrial does not preclude that they could be found in the same formations, climates or habitats (very much alike rodents/lizards and bats/birds can be found and are found in the deposits).

Yet, lagerpetids and pterosaurs are rarely found in the same localities, even in those that have been extensively sampled for years with high preservation potential for both (e.g., microfossil sites in the US southwest). This paper explores a potential reason for this pattern by linking the distribution of these groups to palaeoclimate influence. We fully accept it may not be the only reason. This misunderstanding may arise from the connotation we give to the concept of "palaeoclimate niche" in this paper: we refer to niche with an explicit geographic connotation as defined by the climate indices rather than a theoretical one derived from dietary considerations or from locomotory/reproductive strategy or others (see answers to the comments of Reviewer 1). This is explicitly stated in the methods section pertaining to the quantification of the palaeoclimate niches for each taxonomic group.

The authors do not demonstrate that their results are not an artifact of sampling and fossil record biases – a known and major confounder for paleogeographical studies. They do not include any sort of sampling bias metric in their models and only briefly acknowledge the possibility that their results are influenced by "severe sampling biases". Given the paucity of early pterosaurs, sampling within the basal-most portion of the pterosaur tree is going to be sparse, as noted by the authors, which will inherently affect interpretations of biogeography. Similarly, lagerpetids are a relatively small and species-poor clade, perhaps also indicative of fossil record biases. Proxies of fossil record bias should be included in the analyses.

A. A figure has been included to illustrate the temporal and spatial extent of the fossil data used in our study. Like all other fossil studies, the data is influenced by a number of different factors, which create biases. The dataset used here for pterosauroforms is the most comprehensive available at this time. Yet, it is still too small to compute meaningful

'standardised' species-richness estimates (e.g., through methods such as SQS). In order to avoid introducing further methodological biases, we chose instead to focus on expanding our dataset to be as comprehensive as possible (see newly added text in Supplementary Information). From the new figure (Fig. S1), it is clear that the pterosauro-morph fossil record is restricted. However, note that this does not necessarily imply strong biases, but may be a genuine biological signal (i.e., pterosaurs and lagerpetids may have been rare members of their assemblages). Many of the sites where lagerpetids (and some pterosaurs) are found are microfossil sites (particularly from the southwestern USA) with the potential to preserve fragile fossils. This is a strong mitigation against the sampling biases in those sites: pterosaur materials would likely be found, if present, in these sites (and they have been found in some after extensive sampling: e.g., Petrified Forest National Park). These localities have high preservation potential, particularly for teeth (teeth), but also other fragile fragments (limb bones, delicate lepidosauromorph and fish jaws). Indeed the rarity of pterosaur remains from these assemblages speaks in favour of the hypothesis that these animals were exceedingly rare. Similarly, we broadly discussed the absence of pterosaurs (and lagerpetids) from other sites of exceptional preservations (last paragraph of the Discussion). Their absence from some of these pre-Norian sites is also unlikely to be caused by preservation or sampling biases and may instead be a true biological absence. We further discussed how our strategy mitigates biases in a new section of the Supplementary Information.

Inadequate quantitative results or statistical tests to back up the claims being made: “ Δ -log likelihood” is discussed in the Palaeobiogeography results section, but no numbers are given – no parameter estimates are given anywhere. Δ -log likelihood is usually a measure of the difference in fit between two models. I think that is the case for Biogeobears (though I have never used it). Moreover, there are no quantitative (statistical tests) in this section or the Results section on Ecological Niche Modelling. Qualitative descriptions are inadequate. Is the occurrence of the two groups really “significant” and what parameter values are the models producing under what assumptions?

A. We disagree that we provided a “qualitative description” of the analyses. First, the Δ -log likelihood is indeed a measure of the difference between two dispersal models (in our cases these models are two “distinct rate matrices of dispersal across selected geographical areas” as explained in the method section of “palaeobiogeography”): “The Δ -likelihood is a measure of how strongly the dispersal pattern is supported at any time: in other words, a more negative Δ indicates dispersal/waning of climatic barriers. The Δ measures how much dispersal is occurring by imposing a penalty when the climate barrier is crossed. Thus, the Δ will be more negative when more organisms (=lineages) are dispersing when barriers are present. Conversely, a Δ closer to zero indicates strong barriers to dispersal, or that the crossing of barriers was sporadic and largely followed by cladogenesis.” In other words, the delta-log likelihood is a measure of how much dispersal is occurring when a penalty is imposed. So, because there is a penalty to cross a barrier, the log-likelihood will be more negative when organisms are dispersing when they shouldn't be. From Figure 2A, it looks like there were strong barriers to pterosauro-morph dispersal, and they largely were not crossing barriers, or the crossing of barriers was sporadic and largely followed by cladogenesis and not dispersal across barriers that we believe should be limiting their dispersal. The delta-log is meaningless when observed alone but, as we do, needs to be compared to those of another group (the idea to further report these values is a misunderstanding of the use-case for this analysis). Finally, please refer to the second

paragraph of the section “*Potential of latitudinal dispersion*” and Griffin et al. (2022, Nature, for more details on the methods).

We are confused about the request for statistical tests in this section, as none would be suitable or is indeed necessary. The ENM provides an explicit geographical visualisation of the areas that are quantitatively marked as climatically suitable for each group based on the climatic properties of their distributions and climate models. Conversely, the palaeoclimate niche analyses (i.e. PCA and associated figures) quantitatively test the palaeoclimatic preferences and ranges of pterosaurs and lagerpetids. The results of all statistical tests are reported in the result section of each analysis and in table form in the ED.

Related to the above point, a phylogenetic tree is chosen but doesn't appear to be used in any of the analyses. This is a major potential flaw.

A. The phylogenetic tree is used in the biogeographic analyses, as explicitly stated in the method section “Palaeobiogeography”. The procedure we followed for its assembly is also explained in the corresponding methods section and expanded upon in the Supplementary Information. Using a phylogenetic tree in the other climate-model-based analyses would remove a large amount of data (e.g., multiple occurrences of the same taxon and occurrence of indeterminate taxa), which we deemed unwise. Instead, our strategy allows us to tackle the question from independent points of view in that the biogeography analyses explore the distribution of pterosauiromorphs informed by phylogeny. Conversely, the climate models explore their distribution based on specimen occurrences, which feeds more data into the analyses that they need most. In other words, phylogeny would not be informative in the climate-model analyses, and all occurrence data could not be included in the biogeography (because indeterminate/incomplete taxa could not be placed in the evolutionary tree). We find it reassuring that our results from independent analyses are consistent: the fact that all our disparate analyses converge on the same results mean that the signal is present even when we look for it in many different ways and using different raw data (i.e., phylogenetics vs occurrences).

Specific comments

Abstract

Lines 36-37: “Temporal” would be more appropriate than “stratigraphic” in this context as it is not referring specifically to rock or outcrop, but rather overall disparity in the age of fossils (there are other instances of this as well).

A. We have changed this throughout the text as appropriate.

Lines 36-40: It is stated that there are “stratigraphic and morphological gaps” that separate pterosaurs from lagerpetids, then immediately after it is immediately stated that the groups have “long temporal overlap”. These two statements are opposite one another, so it would clarify things immensely to give some specific geological time frames when they do and do not overlap in time and space.

A. The stratigraphic and morphological gaps between these clades refer to the interval between the Middle to early Upper Triassic (Ezcurra et al. 2020; Foffa et al. 2022 etc.), where only lagerpetids are found, and there is no pterosaur available in the fossil record (no body fossil, no trace fossils), but are inferred through ghost lineages. Yet, pterosaurs and lagerpetids both occur in formations in the rest of the Late Triassic (~middle Norian–Rhaetian).

Lines 49-50: “Spatial segregation and climatic niche distinctions characterized early pterosauiromorph evolution.” This feels very abrupt and is perhaps too definite a statement. I would recommend adding nuance with some sort of lead-in, perhaps, “These findings suggest that spatial segregation...”

A. We have rephrased as suggested.

Main

Lines 86-89: “Lagerpetids are found in continental fluviolacustrine deposits on at least current four continents and maintained a wide latitudinal and geographic spread throughout their evolutionary history (late Ladinian–late Norian, 237–211 Ma; Fig. 1).” Lagerpetids do not go extinct until the end-Triassic (~201 Ma) and are shown in the Rhaetian map in Figure 1.

A. We have rephrased this sentence.

Lines 91-92: “From the upper Norian onward, pterosaurs spread to higher latitudes and in a broader array of habitats”. Figure 1 makes it look like pterosaurs are consistently concentrated at mid latitudes. One taxon, Arcticodactylus, occurs northward in the Rhaetian, but to say that pterosaurs are spreading northwards based on a single taxon feels like an overgeneralization.

A. We remade Figure 1. Hopefully, this new subdivision of the Norian highlights the geographical and stratigraphic distributions of both groups more clearly. See also Fig. S1.

Results

Lines 114-117: “Our model suggests that lagerpetids – and pterosauiromorphs as a whole – likely originated in southwestern Pangaea (i.e., modern South America), whereas the origin of pterosaurs was found at low latitudes in the Northern Hemisphere (ED Fig. 1), consistent with previous studies.” Of course, this is where the model will place the origin of these groups, as this is where all the fossils are. Some nuance should be added to this statement, as these origins are entirely based on the tree and sampling of fossils.

A. We added some nuance to this sentence as requested, and specified that our analyses are phylogeny-informed and refer to a new section in the Supplementary Information that further discusses the limitations of the data. This statement is broadly applicable to any biogeographical (and phylogenetic) analysis of fossil organisms. We do not believe that this should prevent us from conducting hypothesis-generating studies. The purpose of this analysis is not to determine the area of origin of pterosaurs and lagerpetids (as made clear in the final discussion paragraph). The estimation of the ancestral area from phylogeny is a necessary step to determine the dispersal behaviour of these groups.

Lines 159-161: “There are outliers to this pattern, namely Arcticodactylus from Greenland, Caelestiventus from the southwestern USA, and Pachagnathus and Yelaphomte from Argentina”. Four outliers is quite a large percentage of a dataset with only 29 taxa (~14%). Are they really outliers, or is this just another instance of sampling bias?

A. We removed this sentence to avoid confusion. The majority of early pterosaurs are unarguably from mid-low-palaeolatitudes. This may be indeed an instance of sampling bias, although the fact that most of these taxa (except *Arcticodactylus*, whose phylogenetic position remains uncertain) are all deeply nested amongst low-latitude pterosaurs, suggests that their placement may be a genuine signal.

Lines 165-167: "It is possible that these high latitude pterosaur occurrences are evidence of the clade's first dispersal event outside of its hypothesised low-latitude ancestral area (see Discussion)." These pterosaur occurrences are at a lower paleolatitude than the "ancestral area" near the Tethys.

A. We do not understand this comment. Only *Caelestiventus* is found at an equivalent palaeolatitude to the 'ancestral area'... all others are found at higher latitude in the northern hemisphere (*Arcticodactylus*) or higher latitude in the southern hemisphere (*Pachagnathus* and *Yelaphomte*).

Lines 178-180: "Pterosaur potential distribution appears overall patchier and restricted latitudinally, spreading out only from the Carnian, compared to the more homogeneous and latitudinally broader distribution of lagerpetids." According to Figure 1 and the supplementary data, pterosaurs actually have a broader latitudinal range than lagerpetids by about 10 degrees (Lagerpetids: -49.13 to 34.8 paleolat; Pterosaurs: -47.18 to 43.91 paleolat).

A. In light of the new models, we rephrased this sentence. Here, we refer to the habitat suitability in Figure 4 (and ED Figure 4), where it is clear that lagerpetid habitable space is more homogeneous and continuous.

Discussion

Lines 230-235: "The preference of pterosaurs for wet, tropical low latitude areas before the Norian may imply a narrower, more specialised climatic preference, and/or that these suitable conditions were initially limited to equatorial areas. Nevertheless, suitable conditions for pterosaurs are found globally in the upper Norian–Rhaetian (Fig. 4b,d), in accordance with the pattern of quantified latitudinal dispersal and increased geographical spread in the fossil record (Fig. 1)." If there are plenty of "suitable conditions" found globally for pterosaurs in the Norian, but they are only found in the Tethys region, is that not very strong evidence for sampling bias?

A. It is a possibility, but not necessarily the only one, especially considering the many well-sampled formations that, although estimated at highly suitable, do not yield pterosaur fossils. The palaeoclimate conditions may have been suitable in certain areas, but the group may have been historically precluded from reaching those suitable areas. Or they may have been suitable at the "wrong time" or "place". We do not expect a group to be present in all areas of high suitable climate as many other factors might be in play (the same way there are no polar bears in Antarctica or penguins in Greenland/Arctic Canada or Russia).

Lines 266-267: "However, it is worth noting that while there is likely a genuine biogeographical signal in these observations, they could also reflect severe sampling biases". This is the first time sampling bias is mentioned. If it is acknowledged that there could be severe sampling biases, why are they not accounted for in the model?

A. We further discuss the fossil record of pterosauriforms and the impacts of sampling heterogeneity in a new section in the Supplementary Information. We also expand on this point in the next response below.

Methods

Given the statement about there being severe sampling biases, I would like to see more investigation into how such biases impact the results and conclusions reported here. I recommend an approach outlined by Dunne et al. (2021), which standardized fossil

collections with geographic space and Gardner et al. 2019, which studied how fossil record biases affect dispersal estimates. Otherwise, a compelling reason to exclude such analyses should be stated.

A. Unfortunately the same approaches cannot be applied to the data we use in this study. Gardner et al. (2019) used a sampling-standardised approach to specifically examine dispersal, which is not a focus of our study. However, more importantly, Gardner et al. (2019) used formation counts as a proxy for sampling effort, which has been shown to correlate with species richness and is therefore not recommended as a proxy for sampling (see Dunhill et al. 2018, *Palaeontology*). Dunne et al. (2021) used a method based on coverage-based rarefaction (similar to Shareholder Quorum Subsampling; SQS) with the purpose of estimating changes in latitudinal diversity. Such methods require larger amounts of data to compute estimates for species-richness, and so are not suitable for the dataset used in our study. Furthermore, these methods are being overtaken by approaches that specifically compute spatial subsampling, though these new methods, too, require larger amounts of data than are available for pterosauroforms. Within these constraints, we used multivariate approaches that quantitatively investigate patterns within the available data. In particular, the habitat suitability modelling attempts to mitigate biases using a probabilistic approach that interpolates from fragmentary occurrences to create a probabilistic geographical range (= the areas of high suitability).

A more detailed justification for the tree used in the model should be included. The ability of the model to infer ancestral locations and dispersal rates is entirely dependent on the topology of the tree, especially for poorly sampled groups such as early pterosauroforms. I recognize that no tree will ever be perfect, but I think that a discussion of the various trees that exist and potential points of ambiguity is warranted. As of now, the methods simply state that analyses are based on "...a randomly selected most parsimonious tree of Archosauromorpha...(line 356)". Do the results change when a different most parsimonious tree is selected? How many most parsimonious trees exist? I would argue it would also be good to run an analysis using a strict consensus, removing as much ambiguity as possible, and seeing how that impacts results.

A. We have explained this in detail in a new paragraph added to the Supplementary Information. Here's an extended explanation for our tree choice.

Griffin et al. (2022) demonstrated that the choice of tree does not affect the patterns in Dinosauria. The main issue here concerns pterosauroforms that were less sampled in Griffin et al. (2022). The pterosaur tree is well resolved and essentially corresponds to a strict consensus (SC) topology. That said, we could not use a SC tree for lagerpetids because there is no well-resolved SC or even a partial consensus tree. This in itself makes the choice of a random tree the most 'fair'.

However, we notice that the geographical distribution of basal lagerpetid taxa is stable, and the uncertainties occur amongst taxa of broadly similar regions (Argentina or Brazil). This does not affect the results of the biogeographic analyses. In other words, there may be a variation in step orders in the middle and terminal part of the lagerpetid tree, but the succession of ancestral states (areas of origins in our case) at the tips will remain broadly the same. A South American taxon (*Faxinalipterus*) is stable at the root of the clade, followed by *Scleromochlus* (Europe). Further up the order of the other taxa doesn't particularly matter because there will always be a minority of younger North American taxa (*Dromomeron romeri* + *D. gregorii*) within a broader sample of contemporaneous and older South American/southern hemisphere taxa.

Regarding pterosaurs, *Arcticodactylus* and MCSNB 8950 are not included in the phylogenetic tree. We argue that the inclusion of *Arcticodactylus* and MCSNB 8950 would not significantly alter the patterns we recovered in our biogeographical analyses. MCSNB 8950 from northern Italy would cause limited changes in the results of the palaeobiogeography-based analyses. Including *Arcticodactylus* would also not significantly alter the palaeobiogeographic analyses of the potential to dispersion (because of its phylogenetic position and geographical binning) and would simply exaggerate the initial peak of pterosaur dispersion (Fig. 2b–c), leaving the overall pattern otherwise unchanged. However, we still decided against grafting these taxa into the tree because their phylogenetic positions are too uncertain, and they are both in need of revision.

Figures

Figure 1: Despite limited sympatry of pterosaurs and lagerpetids being a central point of this study, Figure 1 makes it look like they almost always co-occur, and no statistics tests are given one way or another. I would recommend modifying the presentation of data, perhaps with zoomed in panels so that it is clear they are not occurring in the same localities. It is also worth considering breaking the maps up into finer temporal resolution. Changes in biogeography within stages are frequently discussed, so a maps for entire stages may be misleading. For instance, with lines 89-91: “Conversely, the earliest body fossils of unambiguous pterosaur are middle Norian in age and found only in a series of marine strata in a low latitude area bordering the Tethys Ocean.” I was initially confused as additional pterosaur occurrences are shown in Figure 1b in North America, far away from the Tethys. I understand after reading in more detail that the North American pterosaurs are Late Norian, versus the middle Norian taxa near the Tethys, but to avoid any misinterpretation I think some sort of additional temporal component needs to be added to this figure. Perhaps adding an additional color scale to the points for early, middle, and late Norian (same for Landinian, Carnian, Rhaetian) could add a layer of clarity.

A. We remade Figure 1 at a finer temporal resolution as suggested, and we also added zoomed-in panels to capture the geographical resolution for the Late Triassic occurrences in North America. We also modified figures 3 and 4 accordingly.

Extended data

Extended Data Figure 1: Faxinalipterus minimus is tip dated to ~215 Ma, but the most recent study for this taxon provides a U-Pb date of 225.42 for the locality that produced it (Kellner et al., 2022). Given that it is seemingly a basal taxon, this younger date could be greatly impacting model results. However, even if the date of 225 is more accurate, it is still relatively geologically younger than other lagerpetids. When relatively late-occurring species in a clade are recovered in very basal positions, it’s worth examining the possible causes, and asking if that position is realistic or possibly an artifact of some biases. One explanation might be a genuine ghost lineage. Other causes might include very incomplete specimens which are minimally comparable to other OTUs in terms of character representation. Further, it is very well documented now that immature archosaurs will often be recovered in basal phylogenetic positions due to expression of ancestral or basal features early in growth. The very small size of the holotype of Faxinalipterus minimus (giving rise to the species name itself) urges some caution that this may be an immature individual and that its very basal position is an artifact of this immaturity. I know this paper is not intended to re-evaluate every taxon included, but in these biogeography models, tree topology and internal node structure make all the difference when it comes to inferring ancestral node locations and number of

dispersal events. If the topology is affected by biases such as ontogeny or specimen incompleteness, the biogeographic results will be affected. I would strongly encourage some nuanced discussion about how biases such as incomplete species and artificial basal position inferred for immature individuals might affect the results of the analyses that depend on tree topology.

A. We changed the tip-dating of *Faxinalipterus* accordingly and re-ran the analyses. The position of immature taxa may indeed affect their position in a phylogenetic analysis. Size, however, is not a good indicator of maturity in early pterosaurs, and we are not aware of any other indication that *Faxinalipterus* may be an immature individual besides its size. However, even if it was, we cannot predict where a mature individual of the same taxon would cluster. So we maintain the position of *Faxinalipterus* where it is. As recommended, we added a discussion on how these biases may affect the results of the analyses in the Supplementary Information and re-ran all the analyses with a revised date for *Faxinalipterus* (with very limited changes from the previous version). We also remade all figures affected by the incorrect dating of *Faxinalipterus*.

Minor edits

Title: I would recommend using the British English spelling of “palaeobiogeography” for consistency. I would further recommend checking for consistency in the use of British English throughout the manuscript as there are several instances of American English (e.g., characterized instead of characterised).

A. Done.

Line 52: “palaeogeography”. Typo, should be “palaeobiogeography”

A. Done.

Line 60: “>60 million years”. I would recommend spelling out “more than”

A. Done.

Lines 63-64: “The fossil records of pterosaurs and their kins...”. Pluralization. I would recommend “The fossil record of pterosaurs and their kin...”

A. Done.

Line 70: “This work led to the realisation that lagerpetids as the closest relatives...” Should be “This work led to the realisation that lagerpetids are the closest relatives...”

A. Done.

References

*Dunne, E.M., Farnsworth, A., Greene, S.E., Lunt, D.J. and Butler, R.J., 2021. Climatic drivers of latitudinal variation in Late Triassic tetrapod diversity. *Palaeontology*, 64(1), pp.101-117.*

*Gardner, J., K. Surya, and C. L. Organ (2019). Early tetrapodomorph biogeography: controlling for fossil record bias in macroevolutionary analyses. *Comptes Rendus Palevol*. 18 (7): 693-908. doi.org/10.1016/j.crpv.2019.10.008.*

*Kellner, A.W., Holgado, B., Grillo, O., Pretto, F.A., Kerber, L., Pinheiro, F.L., Soares, M.B., Schultz, C.L., Lopes, R.T., Araújo, O. and Müller, R.T., 2022. Reassessment of *Faxinalipterus minimus*, a purported Triassic pterosaur from southern Brazil with the description of a new taxon. *PeerJ*, 10, p.e13276.*

Reviewer #3: pterosaurs (Remarks to the Author):

To investigate early pterosauiromorph distribution in time and space plus environmental data, this work assumes (as modern consensus) that lagerpetids are the closest relatives of pterosaurs and thus, uniting the two groups in the Pterosauiromorpha, with the Lagerpetidae as the non-flighted forms, whereas pterosaurs are the potentially active flyers. The work, therefore, offers an overview of this distribution based on this assumption, which leaves me curious about how these results might have appeared if another group had been considered instead. But within this scope, I find that the authors effectively handle the results of their analyses and arrive at reasonable, well-funded conclusions.

Regarding the arboreal ecomorphology assumption to early pterosaurs, arboreality has been assumed for some taxa and related to the origin of pterosaur flight, but we still lack quantitative ecomorphological analyses to support this hypothesis. Nevertheless, based on the role that arboreality likely played in the origin of the group, this hypothesis to explain the more constrained distribution for pterosaurs against the lagerpetids' broader distribution and ecological versatility seems plausible.

Therefore, overall, it can be said that the work is quite well done, meticulously crafted with hypotheses grounded in robust and well-justified analyses.

A. We thank the reviewer for the kind words, and we hope they'll agree that the changes we made improved the manuscript.

As minor comments:

- at Fig. 1, the originally described material of Faxinalipterus (not Faxinalpterus as plotted in Fig. 1b) was disassociated into a new putative pterosauiromorph, Maehary bonapartei by Kellner et al. (2022), which was considered the earliest-diverging member of Pterosauiromorpha. As the figure shows the Triassic pterosauiromorph occurrences, it would be nice to include Maehary overlapping Faxinalipterus at the same spot (to improve this diversity along time). This reference was mentioned by the authors, but was Maehary indeed considered in this study? (if not, shouldn't it?)

A. Based on the most recent phylogenetic analyses, *Maehary* is not considered a pterosauiromorph anymore (but is recovered as a pseudosuchian gracilisuchid) by Müller et al. (2023) and subsequent analyses. We agree with this result, as we also did not find support for pterosauiromorph affinity for *Maehary* in a different study (Foffa et al. 2023). Based on these lines of evidence we decided against including this taxon in the study. Nevertheless, note that including this taxon would be unlikely to change any of the results of our analyses because of its supposed phylogenetic position at the base of Pterosauiromorpha and the age and palaeogeography of its site of origin.

-data of climate niche analysis is empty at figshare; link of climatic suitability analysis (in Data availability) is also missing.

A. We are sorry this link did not work. The data were also attached as supplementary information as well. Hopefully, all data and codes are accessible now.

-at the table: there is a double PULR 06 (number) referred to Lagerpeton chanarensis to be checked (as mentioned by the authors); was this checked? It seems to have been something that the authors missed.

A. Thank you for pointing this out. We corrected the mistake and deleted the double line as well as corrected some other similar small mistakes and updated the age/occurrences of

some entries. We updated the spreadsheet including more precise information and references.

Dear Editor and reviewers,

We thank you and the reviewers for the comments on the revised submission. We are pleased that our work on the manuscript was well received, and we are delighted to receive such positive revisions.

Please find below a point-by-point discussion of the few remaining points highlighted in the second round of review.

We are delighted to resubmit this work, and we hope that this revised submission is considered suitable for publication.

Davide Foffa (on behalf of the authors)

Reviewer #1:

Remarks to the Author:

I congratulate the authors for both the careful responses to the comments of the reviewers (including mine) and the extensive revisions to the manuscript. I am particularly impressed with the new text added to both the main text portion of the manuscript and the supplementary information regarding the palaeoclimate, the computer modeling and the geologic evidence for the palaeoclimate.

A few points I would like to your attention:

RESPONSE LETTER

You stated in your letter "we lack the data to investigate whether dietary/habitat/reproductive/physiological aspects were contributing factors." This would certainly be an interesting line for future investigation.

A. Agreed. Let's hope there will be soon enough data to open this line of research.

MAIN TEXT

There seems to be some confusion as to the age of the earliest pterosaur fossils. In lines 238-240, the oldest known pterosaur fossils are described as middle Norian in age, including both the northern Tethyan region and the Chinle-Dockum basins. The problem is that in lines 538-542 you cite pterosaur finds from the Tecovas Fm (Dockum) and Blue Mesa Mbr (Chinle), both of which have possible early Norian ages. Please clarify.

A. We clarified these sections. The earliest fossils referable to pterosaurs are indeed the Chinle/Dockum ones in the early-middle Norian.

SUPPLEMENTARY INFORMATION

Same issue appears here in lines 32-34 where you describe the earliest pterosaurs as middle Norian and exclusively from the Tethyan realm. What about the Chinle-Dockum fossils?

A. See comment above.

lines 170-173. I agree absolutely with this approach. The relationship between orbital cyclicity and climate is complex and the cycle periods are not constant through geologic time (as commonly supposed). The effects of the cycles on insolation are typically small and

VERY latitude dependent. Given cycle periods of 10s ky to 100s ky, time averaging insolation is the only practical solution to modeling climate for specific times.

A. Great!

lines 181-183. Foster et al. (2017) is not a bad source for palaeo-pCO₂. The best thing about this study is that it presents the VERY diverging results from the stomatal and soil carbonate proxies. I could point out that there are more recent (although not necessarily better) models of Triassic pCO₂, but it would not be reasonable to ask you to rerun the model just to use slightly different input.

A. Glad that our approach is considered suitable. We agree that it would be interesting to see future work based on models with different pCO₂ sources.

Remarks on code availability:

I am sorry to say that examination and evaluation of the code used for this research is beyond my area of expertise.

Reviewer #3:

Remarks to the Author:

I am satisfied with the arguments presented and commend the authors for their effort in providing well-supported justifications in response to the issues raised during the review process. I believe the manuscript has the potential to significantly contribute to the ongoing debate regarding the palaeogeography of lagerpetids and pterosaurs.

A. Thank you!

Remarks on code availability:

At this point, I was able to open all available files with the main manuscript.

A. Excellent.